# Differential regulation of hepatic physiology and injury by the TAM receptors Axl and Mer

Anna Zagórska[1],*, Paqui G Través[1,2],*, Lidia Jiménez-García[1], Jenna D Strickland[3], Joanne Oh[1], Francisco J Tapia[1], Rafael Mayoral[4], Patrick Burrola[1], Bryan L Copple[3], Greg Lemke[1,5]

Genome-wide association studies have implicated the TAM receptor tyrosine kinase (RTK) Mer in liver disease, yet our understanding of the role that Mer and its related RTKs Tyro3 and Axl play in liver homeostasis and the response to acute injury is limited. We find that Mer and Axl are most prominently expressed in hepatic Kupffer and endothelial cells and that as mice lacking these RTKs age, they develop profound liver disease characterized by apoptotic cell accumulation and immune activation. We further find that Mer is critical to the phagocytosis of apoptotic hepatocytes generated in settings of acute hepatic injury, and that Mer and Axl act in concert to inhibit cytokine production in these settings. In contrast, we find that Axl is uniquely important in mitigating liver damage during acetaminophen intoxication. Although Mer and Axl are protective in acute injury models, we find that Axl exacerbates fibrosis in a model of chronic injury. These divergent effects have important implications for the design and implementation of TAM-directed therapeutics that might target these RTKs in the liver.

## Introduction

Liver diseases—including acute liver failure, viral hepatitis, and alcoholic and nonalcoholic fatty liver disease (NAFLD)—represent a major medical burden worldwide (Stravitz & Kramer, 2009; Corey & Kaplan, 2014; Louvet & Mathurin, 2015). Increasing evidence suggests that both progression and resolution of these diseases depend on the kinetics and intensity of innate and adaptive immune responses (Sipeki et al, 2014; Guidotti et al, 2015) and that macrophages—including Kupffer cells (KCs), the resident macrophages of the liver—are important regulation loci (Smith, 2013).

We have shown that the TAM receptor tyrosine kinases (RTKs)—Tyro3, Axl, and Mer (Lemke, 2013)—are pivotal modulators of tissue macrophage function generally (Lu & Lemke, 2001; Rothlin et al, 2007; Zagórska et al, 2014; Dransfield et al, 2015; Fourgeaud et al, 2016; Lemke, 2019). Over the last several years, genome-wide association studies have tied polymorphisms in the human *MERTK* gene—encoding Mer—to altered risk for both (a) fibrosis in patients with chronic hepatitis C virus infection (Patin et al, 2012; Rueger et al, 2014; Matsuura & Tanaka, 2016; Jimenez-Sousa et al, 2018) and (b) NAFLD, in which two intronic single-nucleotide *MERTK* polymorphisms are protective (Petta et al, 2016; Musso et al, 2017). In the progression from NAFLD to nonalcoholic steatohepatitis (NASH), these polymorphisms, which are associated with *reduced* Mer expression, are linked to reduced risk for liver fibrosis (Cavalli et al, 2017). In turn, recent analyses have indicated that *Mertk*$^{-/-}$ mice display reduced levels of a NASH-like fibrosis that is induced by high-fat diet, via reduced activation of hepatic stellate cells by macrophages that are normally Mer$^+$ (Cai et al, 2019). Together, these findings suggest that Mer signaling promotes hepatic fibrosis. Independently, patients with acute liver failure have been found to display markedly elevated numbers of Mer$^+$ macrophages and monocytes in their liver, lymph nodes, and circulation (Barcena et al, 2015; Bernsmeier et al, 2015; Triantafyllou et al, 2018), and Mer has, therefore, emerged as a target in the treatment of liver disease (Mukherjee et al, 2016; Bellan et al, 2019).

With respect to Axl, elevated serum levels of soluble Axl extracellular domain (sAxl) have been found to be a biomarker for hepatocellular carcinoma (Reichl & Mikulits, 2016), and mice lacking Gas6, the obligate Axl ligand (Lew et al, 2014), display enhanced tissue damage in a liver ischemia model (Llacuna et al, 2010). At the same time, Axl$^+$ monocytes are elevated in patients with cirrhosis (Brenig et al, 2020), and serum Gas6 and sAxl levels are elevated in patients with hepatitis C virus and alcoholic liver disease (Barcena et al, 2015). Divergent roles for Axl and Mer have been reported in chronic models of fibrosis, where *Mertk*$^{-/-}$ mice exhibited enhanced NASH development when fed a high-fat diet, whereas *Axl*$^{-/-}$ mice were protected (Tutusaus et al, 2019). These

[1]Molecular Neurobiology Laboratory, The Salk Institute, La Jolla, CA, USA    [2]Instituto de Investigaciones Biomédicas Alberto Sols (CSIC-UAM), Madrid, Spain    [3]Department of Pharmacology & Toxicology, Michigan State University, East Lansing, MI, USA    [4]Division of Endocrinology & Metabolism, Department of Medicine, University of California, San Diego, La Jolla, CA, USA    [5]Immunobiology and Microbial Pathogenesis Laboratory, The Salk Institute, La Jolla, CA, USA

Correspondence: lemke@salk.edu
*Anna Zagórska and Paqui G Través contributed equally to this work

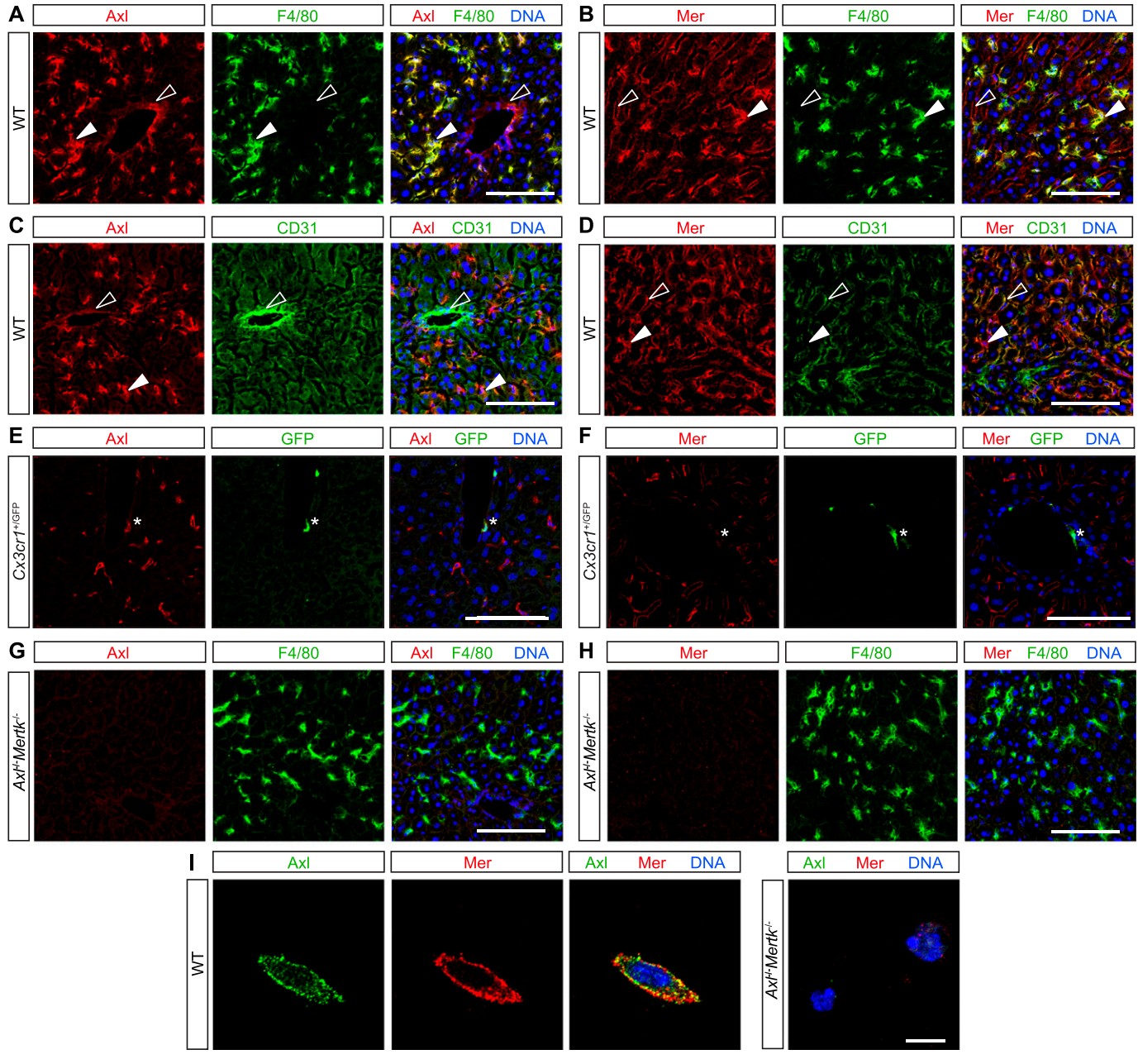

**Figure 1. TAM expression in normal liver.**
**(A, B, C, D, E, F, G, H)** Mouse liver immunohistochemistry. **(A, C, E)** Axl (red) is expressed by F4/80+ Kupffer cells (KCs) (green, closed arrowheads) and strongly CD31+ blood-vessel-lining endothelial cells (green, open arrowheads) and by perivascular macrophages (E, green, asterisk). **(B, D, F)** Mer (red) is expressed by KCs (green, closed arrowheads), by weakly CD31+ sinus-lining endothelial cells (green, open arrowheads), and very weakly by perivascular macrophages (F, green, asterisk). **(G, H)** Mertk−/−Axl−/− mice were used for antibody controls. **(I)** Basal TAM expression on isolated KCs. Axl and Mer are co-expressed on the surface of individual cultured KCs. (See the Materials and Methods section for KC purification and culture). Bars (A, B, C, D, E, F, G, H): 100 μm, and (I): 10 μm.

multiple findings notwithstanding, the general importance of TAM receptor signaling to both normal liver physiology and to acute, rapid-onset liver insults has not been assessed. We have, therefore, exploited a set of conventional and conditional mouse mutants in the *Axl* and *Mertk* genes and subjected these mutants to established models of both acute liver damage and chronic fibrosis, to make these assessments.

# Results

## Expression of TAM receptors in mouse liver

We first used immunohistochemistry (IHC) to delineate TAM expression in adult mouse liver. Most prominently, we detected very strong expression of both Axl (Fig 1A) and Mer (Fig 1B) in all KCs.

These liver macrophages did not express detectable Tyro3 (data not shown). Most tissue macrophages (e.g., peritoneal macrophages and microglia) express high levels of Mer and low levels of Axl at steady state (Zagórska et al, 2014; ImmGen, 2016), and so KCs fall into the restricted subset of unusual macrophages, including red pulp macrophages of the spleen (Lu & Lemke, 2001; ImmGen, 2016) and alveolar macrophages of the lungs (Fujimori et al, 2015; ImmGen, 2016), that abundantly express both receptors. Axl and Mer were also expressed in many CD31$^+$ endothelial cells (ECs) of the liver vasculature, with Axl strongly in CD31$^+$ blood vessels (Fig 1C) and Mer weakly in CD31$^+$ hepatic sinuses (Fig 1D). Axl was also expressed in perivascular macrophages, which only weakly expressed Mer (Fig 1E and F). Antibody specificity was confirmed using $Axl^{-/-}Mertk^{-/-}$ tissues and cells (Fig 1G and H). Axl and Mer were co-expressed on freshly isolated KCs (Fig 1I).

## Role of TAM receptors in hepatic aging

Given these expression data, we asked if Axl and Mer were relevant to normal hepatic aging by examining the livers of aged (8–12 mo) $Axl^{-/-}Mertk^{-/-}$ mice versus wild-type (WT) mice. Remarkably, we found that multiple inflammation and tissue damage markers were markedly elevated in the aged $Axl^{-/-}Mertk^{-/-}$ liver, in the absence of any overt perturbation (Fig 2). Cleaved caspase 3$^+$ (cCasp3$^+$) apoptotic cells (ACs) were elevated 10-fold (Fig 2A), consistent with the essential role that TAM receptors play in AC clearance (Lemke & Burstyn-Cohen, 2010; Lew et al, 2014; Zagórska et al, 2014; Lemke, 2019). Liver expression of the scavenger receptor MARCO, an indicator of macrophage activation, was dramatically higher in the double mutants (Fig 2B), as was GFP expression in $Cx3cr1^{GFP/+}Axl^{-/-}Mertk^{-/-}$ mice, indicative of the immune infiltration of CX3CR1$^+$ monocytes (Fig 2C). The general histology of the aged $Axl^{-/-}Mertk^{-/-}$ liver, as assessed by hematoxylin and eosin staining, was marked by a pronounced increase in immune infiltrates relative to the wild-type liver (Fig S1A). This enhanced cellularity was most prominent in the liver parenchyma surrounding large blood vessels (Fig S1A).

Liver mRNAs encoding multiple proinflammatory cytokines, immune modulators, and chemokines were elevated 4- to 20-fold in aged $Axl^{-/-}Mertk^{-/-}$ mice compared with WT (Fig 2D), and the serum levels of two enzymes associated with liver damage—alanine and aspartate aminotransferase (ALT and AST, respectively)—were similarly elevated (Fig 2E). When we examined the expression of cytokine and chemokine mRNAs in slightly younger (6–8 mo) animals, we again measured elevated levels of these mRNAs in $Axl^{-/-}Mertk^{-/-}$ mice versus wild-type (WT) mice (Fig S1B–F). The extent of elevation was, in general, slightly reduced relative to 8–12-mo mice (Figs 2D and S1B–F), consistent with a phenotype that worsens with increasing age. Although certain cytokine and chemokine mRNAs were also elevated in the livers of $Axl^{-/-}$ and $Mertk^{-/-}$ single mutants at 6–8 mo, robust elevation in general required genetic inactivation of both receptors (Fig S1B–F). Together, these results demonstrated that the aged $Axl^{-/-}Mertk^{-/-}$ livers were damaged and inflamed in the absence of any experimental insult and that TAM signaling is, therefore, required for normal liver homeostasis and healthy aging.

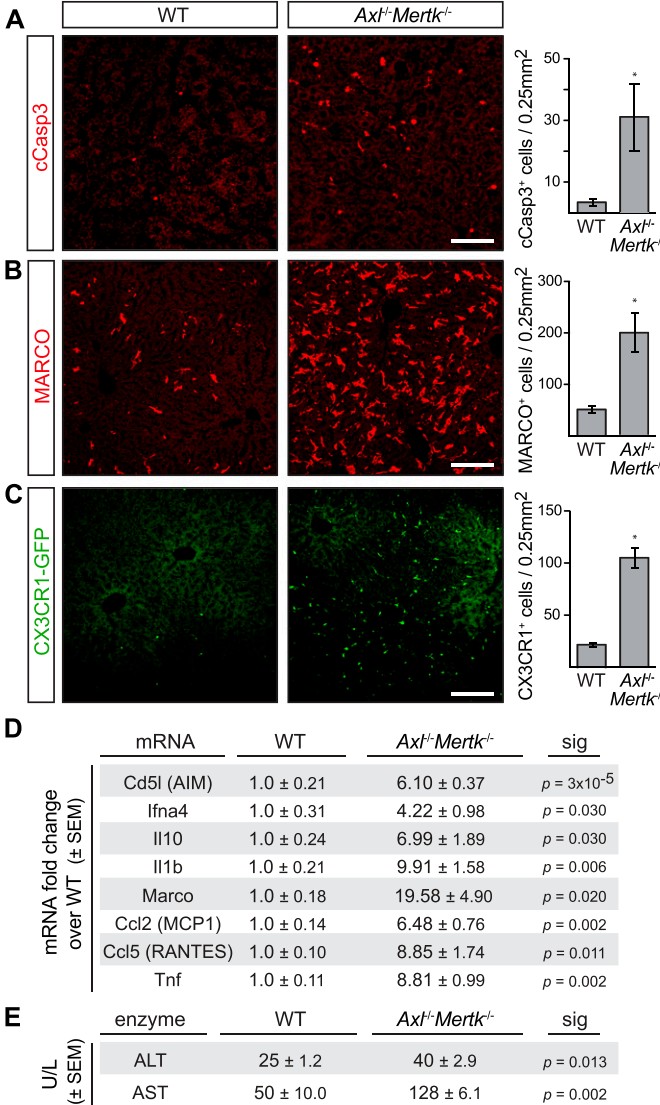

**Figure 2. Liver pathology in aged $Axl^{-/-}Mertk^{-/-}$ mice.**
**(A)** 9–12-mo-old $Axl^{-/-}Mertk^{-/-}$ mice accumulate approximately ninefold more cCasp3$^+$ apoptotic cells than WT mice. **(B)** 9–12-mo-old $Axl^{-/-}Mertk^{-/-}$ exhibit an approximately fourfold increase in MARCO staining in liver than WT. **(C)** 9–12-mo-old $Axl^{-/-}Mertk^{-/-}Cx3Cr1^{+/GFP}$ mice have approximately fivefold more infiltrating GFP$^+$ immune cells in liver than $Cx3Cr1^{+/GFP}$. **(D)** Levels of the indicated mRNAs, isolated from 8 to 12 mo livers, and quantified by qRT-PCR relative to WT. **(E)** Serum ALT and AST activity from 7-mo-old WT and $Axl^{-/-}Mertk^{-/-}$ mice. **(A, B, C)** Representative images from 3 to 4 mice per genotype. **(D, E)** Representative data from two independent experiments (n = 3–4 mice per genotype). Bars (A, B, C): 100 $\mu$m; *$P$ < 0.05. $t$ test.

**D**

| mRNA | WT | $Axl^{-/-}Mertk^{-/-}$ | sig |
|---|---|---|---|
| Cd5l (AIM) | 1.0 ± 0.21 | 6.10 ± 0.37 | $p = 3 \times 10^{-5}$ |
| Ifna4 | 1.0 ± 0.31 | 4.22 ± 0.98 | $p = 0.030$ |
| Il10 | 1.0 ± 0.24 | 6.99 ± 1.89 | $p = 0.030$ |
| Il1b | 1.0 ± 0.21 | 9.91 ± 1.58 | $p = 0.006$ |
| Marco | 1.0 ± 0.18 | 19.58 ± 4.90 | $p = 0.020$ |
| Ccl2 (MCP1) | 1.0 ± 0.14 | 6.48 ± 0.76 | $p = 0.002$ |
| Ccl5 (RANTES) | 1.0 ± 0.10 | 8.85 ± 1.74 | $p = 0.011$ |
| Tnf | 1.0 ± 0.11 | 8.81 ± 0.99 | $p = 0.002$ |

mRNA fold change over WT (± SEM)

**E**

| enzyme | WT | $Axl^{-/-}Mertk^{-/-}$ | sig |
|---|---|---|---|
| ALT | 25 ± 1.2 | 40 ± 2.9 | $p = 0.013$ |
| AST | 50 ± 10.0 | 128 ± 6.1 | $p = 0.002$ |

U/L (± SEM)

## Role of TAM receptors in Jo2 and LPS/D-Gal acute injury models

We next asked how TAM receptor mutants would fare in two acute liver injury models—a fulminant hepatitis model based on injection of the Jo2 anti-Fas antibody (Lacronique et al, 1996), and an endotoxic shock model precipitated by injection of LPS and D-galactosamine (Car et al, 1994). A nonlethal i.p. dose of Jo2 (0.3 mg/kg) did not lead to any change in the appearance of either the WT or $Axl^{-/-}$ liver at 24 h after treatment but produced widespread hemorrhage and

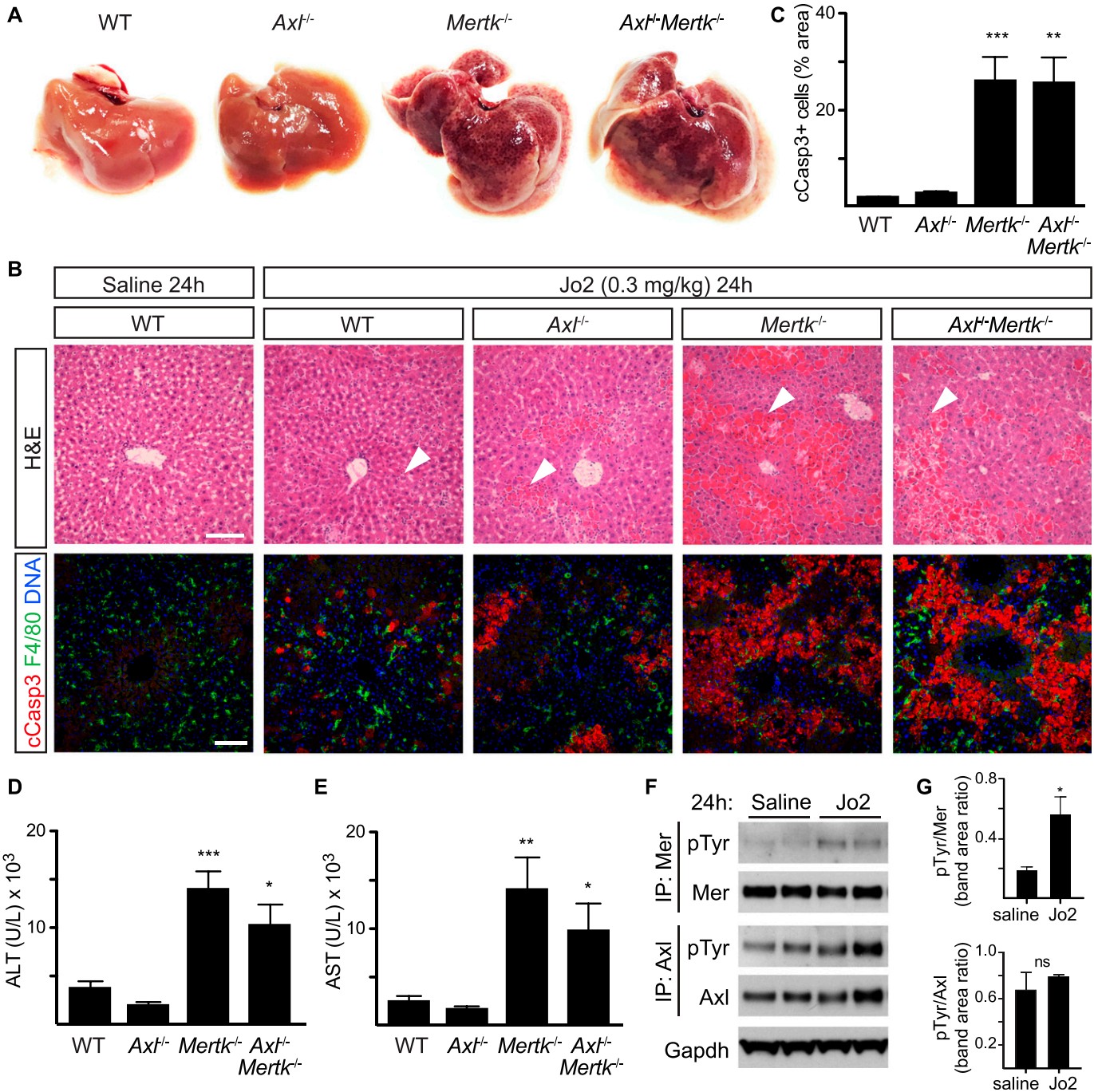

**Figure 3.  Protective role of Mer in nonlethal Jo2 liver damage model.**
**(A)** Extensive hemolysis (mottled darkening) of liver lobes 24 h after a nonlethal (0.3 mg/kg) Jo2 injection in exsanguinated (by cardiac puncture) $Mertk^{-/-}$ and $Axl^{-/-}$ $Mertk^{-/-}$, but not in $WT$ or $Axl^{-/-}$ mice. **(B)** Liver sections were analyzed by H&E staining (top row) and immunohistochemistry with indicated antibodies (bottom row). Markedly increased accumulation of apoptotic cells is observed in $Mertk^{-/-}$ and $Axl^{-/-}Mertk^{-/-}$ liver. Bars: 100 $\mu$m. Arrowheads: damaged tissue and apoptotic cell accumulation. **(A, B)** Representative images from three experiments (n = 3–5 mice per genotype). **(C)** Apoptotic area quantification demonstrates a ~14-fold increase in cCasp3+ cells. $t$ test. **(D, E)** ALT/AST serum activity assays. n = 7–9 mice per genotype. *$P$ < 0.05, **$P$ < 0.01, ***$P$ < 0.001. $t$ test. **(F)** Liver lysates from WT mice injected with saline or Jo2 were immunoprecipitated for Mer and Axl and immunoblotted with indicated antibodies. **(G)** Quantification of the receptor phosphorylation (activation) results in (F). *$P$ < 0.05, $t$ test. Representative image from two experiments (n = 2 mice/treatment).

severe congestion of the sinusoidal space in both the $Mertk^{-/-}$ and $Axl^{-/-}Mertk^{-/-}$ liver (Fig 3A). Similarly, whereas this treatment yielded few uncleared ACs in WT and $Axl^{-/-}$ livers, a 15-fold

increase in ACs was seen in $Mertk^{-/-}$ and $Axl^{-/-}Mertk^{-/-}$ livers (Fig 3B and C). Serum ALT and AST levels were also elevated specifically in $Mertk^{-/-}$ and $Axl^{-/-}Mertk^{-/-}$ mice (Fig 3D and E). Consistent with

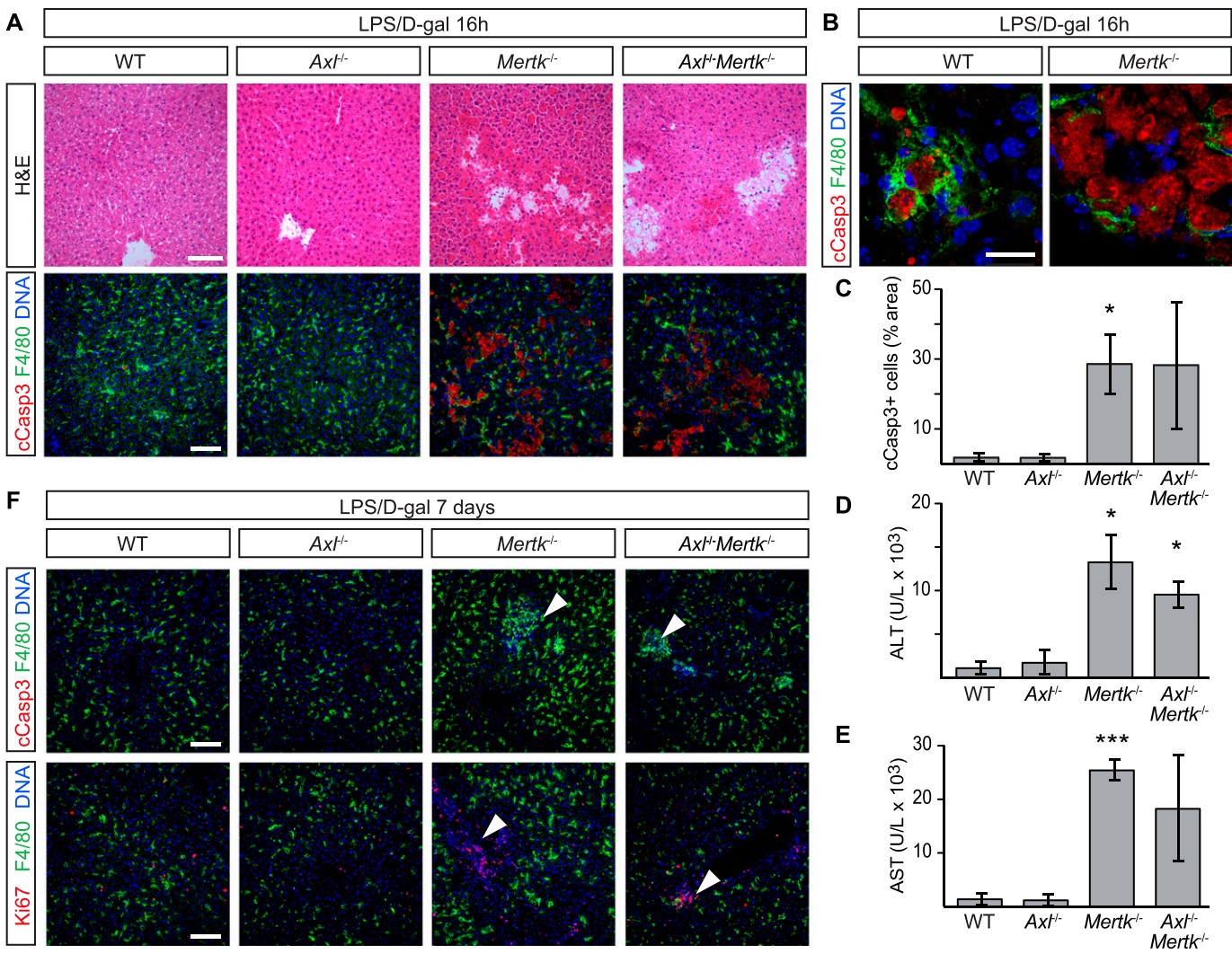

**Figure 4. Protective role of Mer in nonlethal LPS/D-gal liver damage model.**
**(A, B)** Liver sections were analyzed 16 h after i.p. injection with 350 mg/kg D-gal and 1 μg/kg LPS by H&E staining and immunohistochemistry with the indicated antibodies. Increased accumulation of apoptotic cells can be observed in $Mertk^{-/-}$ and $Axl^{-/-}Mertk^{-/-}$. **(A, B)** Bars: 100 μm (A) and 20 μm (B). **(B)** shows that apoptotic cells are engulfed by F4/80$^+$ Kupffer cells. **(C)** Apoptotic area quantification. **(D, E)** ALT and AST activity assays in sera 24 h postinjection. **(F)** Liver sections 7 d postinjection analyzed by H&E staining and immunohistochemistry with the indicated antibodies. Arrowheads in upper and lower rows represent unresolved loci of condensed, necrotic cCasp3$^+$ cells, and proliferating Ki67$^+$ cells, respectively, specifically in $Mertk^{-/-}$ and $Axl^{-/-}Mertk^{-/-}$ livers. Bars: 100 μm. n = 3 per genotype, except $Axl^{-/-}Mertk^{-/-}$ n = 2. $*P \leq 0.05$, $***P \leq 0.005$. $t$ test (C, D, E).

these results, the nonlethal Jo2 dose in WT mice led to enhanced hepatic activation of Mer, but not Axl, as assessed by tyrosine autophosphorylation (Fig 3F and G). When we used a nonlethal dose of LPS/D-gal, we similarly observed very few ACs in WT or $Axl^{-/-}$ mice at 16 h after injection but detected many in both $Mertk^{-/-}$ and $Axl^{-/-}Mertk^{-/-}$ mice (Fig 4A–C). ALT and AST levels were again markedly elevated in $Mertk^{-/-}$ and $Axl^{-/-}Mertk^{-/-}$ mice, but not in $Axl^{-/-}$ mice (Fig 4D and E). Recovery and regeneration from LPS/D-gal damage was delayed in $Mertk^{-/-}$ and $Axl^{-/-}Mertk^{-/-}$ mice because loci of post-apoptotic dead cells and Ki67$^+$ proliferative cells were still present in these mice at 7 d after injection (Fig 4F). Together, these results demonstrated that TAM receptors are required for liver homeostasis and that Mer specifically is essential for the clearance of ACs that are produced during acute injury.

## TAM receptor expression in KCs is critical for liver physiology

We asked whether these phenomena reflected TAM expression in KCs versus ECs. KCs are transiently CX3CR1$^+$ early during their development but are CX3CR1$^-$ in the mature liver (Yona et al, 2013) (Figs 1E and F and 2C), whereas ECs are never CX3CR1$^+$. We used a constitutively active $Cx3cr1^{Cre}$ line, which drives Cre expression early in KC development (Yona et al, 2013), and crossed this line with conditional floxed $Mertk^{f/f}$ (Fourgeaud et al, 2016) and $Axl^{f/f}$ (Schmid et al, 2016) alleles. We also crossed the tamoxifen (Tx)-inducible $Cx3cr1^{CreER/+}$ line (Parkhurst et al, 2013), which we have used previously (Fourgeaud et al, 2016), to these same conditional alleles. The $Cx3cr1^{Cre/+}Axl^{f/f}Mertk^{f/f}$ mice displayed a dramatic reduction in Axl and Mer expression in KCs and peritoneal

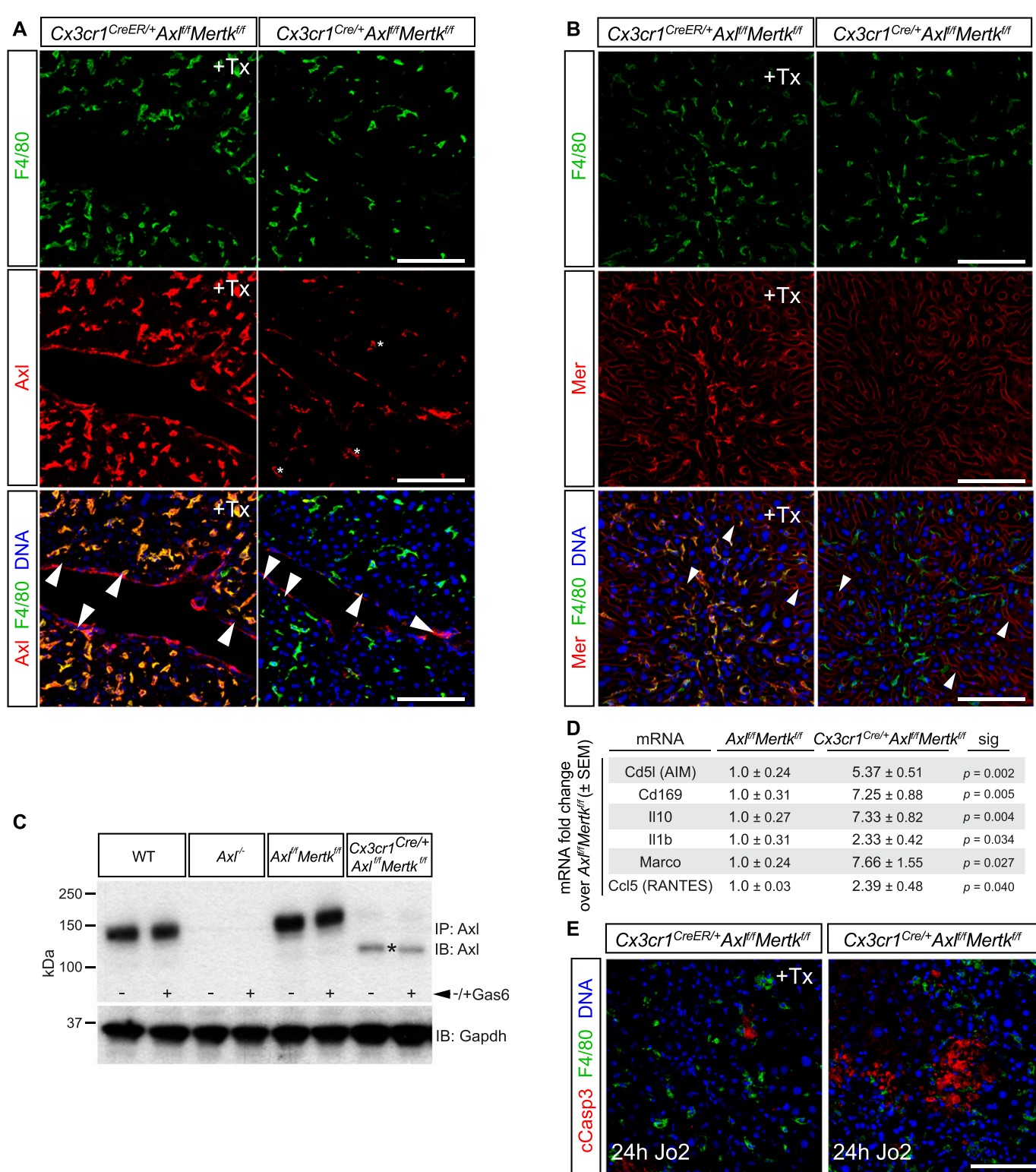

**Figure 5. Axl deletion from Kupffer cells (KCs).**
**(A)** Liver sections from tamoxifen-treated (+Tx) *Cx3cr1^{CreER/+}Axl^{f/f}Mertk^{f/f}* mice and constitutive *Cx3cr1^{Cre/+}Axl^{f/f}Mertk^{f/f}* mice stained with antibodies to F4/80 (top row, green) and Axl (middle row, red), illustrating *Cx3cr1^{Cre/+}*-driven loss of Axl from KCs but not endothelial cells (ECs) in constitutive *Cx3cr1^{Cre/+}Axl^{f/f}Mertk^{f/f}* mice, but normal strong Axl expression in KCs in Tx-treated *Cx3cr1^{CreER/+}Axl^{f/f}Mertk^{f/f}* mice. Bottom row is the merged image of top and middle rows, with DNA visualized by Hoechst 33258 (blue). Note that Axl immunostaining in KCs of the constitutive *Cx3cr1^{Cre/+}Axl^{f/f}Mertk^{f/f}* liver is dramatically reduced but not completely eliminated. Examples of KCs with low residual Axl staining are indicated by asterisks. Axl immunostaining in ECs (arrowheads) persists in both genotypes. **(A, B)** Same analyses as in (A), except that middle row sections are stained with a Mer antibody. Mer immunostaining in ECs (arrowheads) persists in both genotypes. **(C)** Peritoneal macrophages from *Cx3cr1^{Cre/+}Axl^{f/f}Mertk^{f/f}* mice,

macrophages, whereas KC expression in Tx-injected $Cx3cr1^{CreER/+-}$ $Axl^{f/f}Mertk^{f/f}$ mice was unaffected (Fig 5A–C). The $Cx3cr1^{CreER/+}$ line only drives Cre expression in CX3CR1$^+$ cells upon Tx activation, and as noted above, mature KCs are CX3CR1$^-$. At 7 mo of age, the $Cx3cr1^{Cre/+}Axl^{f/f}Mertk^{f/f}$ livers displayed elevated levels of many proinflammatory and immunoregulatory mRNAs relative to $Axl^{f/f}$ $Mertk^{f/f}$ (Fig 5D). In addition, a nonlethal Jo2 dose led to significantly greater AC accumulation in $Cx3cr1^{Cre/+}Axl^{f/f}Mertk^{f/f}$ livers than in Tx-treated $Cx3cr1^{CreER/+}Axl^{f/f}Mertk^{f/f}$ livers (Fig 5E). These results argue that TAM expression in KCs is critical for the phenotypes that develop during both aging and the response to acute injury.

### Role of TAM receptors in acetaminophen-induced acute liver injury

Acetaminophen (APAP)-induced hepatotoxicity is the most frequent cause of acute liver failure worldwide (Bunchorntavakul & Reddy, 2018; Ramachandran & Jaeschke, 2019). We therefore sought to examine the relative performance of WT, $Axl^{-/-}$, and $Mertk^{-/-}$ mouse mutants in a standard model of acute APAP intoxication, which involves overnight (16 h) fasting and subsequent i.p. injection of the drug at 300 mg/kg (see the Materials and Methods section). Although analyses in this model are generally focused on the resolution of inflammation, which occurs between 48 and 72 h after acetaminophen overdose, we found that studies of APAP-treated mice beyond 48 h after drug administration were precluded by the remarkably strong phenotype that we observed specifically in $Axl^{-/-}$ mice. In two independent experimental series, both WT and $Mertk^{-/-}$ mice were motile and superficially normal at 12, 24, and 48 h after APAP, but most $Axl^{-/-}$ mice were very sick and nonmotile across all of this period. Examination of the livers of APAP-treated mice at 48 h after drug administration revealed massive hemorrhage specifically in the $Axl^{-/-}$ mice (Fig 6A and B). A typical $Axl^{-/-}$ APAP-treated liver in situ, seen in 60% of treated mice, is shown in Fig 6A. Histological staining of liver sections 48 h posttreatment revealed substantial hemorrhage within the $Axl^{-/-}$ but not WT or $Mertk^{-/-}$ liver parenchyma (Fig 6B).

Serum levels of ALT were also markedly elevated at 48 h after APAP specifically in $Axl^{-/-}$ relative to WT and $Mertk^{-/-}$ mice (Fig 6C), and although APAP intoxication is primarily associated with necrosis, APAP-treated $Axl^{-/-}$ livers also displayed elevated numbers of cCasp3$^+$ cells relative to wild-type (Fig 6D). Enzymatic activation of the Axl tyrosine kinase always triggers downstream metalloprotease cleavage of the Axl extracellular domain from the cell surface and the generation of soluble Axl (sAxl) (Lemke, 2013; Zagórska et al, 2014; Orme et al, 2016). Correspondingly, we measured elevation of circulating sAxl in serum 48 h after APAP treatment of WT mice (Fig 6E). The oxidative stress that is induced by APAP is known to induce Axl expression and activation in multiple cell types (Konishi et al, 2004; Huang et al, 2013), and so the elevated

sAxl in serum may be related to this induction. We did not detect significant elevation of $Cd5l$, $Il10$, $Il1b$, $Ccl2$, $Ccl5$, or $Tnf$ mRNAs in $Axl^{-/-}$ versus either WT or $Mertk^{-/-}$ livers, indicating that the Axl-specific liver damage induced by APAP was not due to specific elevation of these cytokine/chemokine mRNAs (Fig 6F). Consistent with our results, earlier APAP studies documented only modest (1.4-fold) increases in necrosis and neutrophil infiltration in the liver of $Mertk^{-/-}$ mice at 8 h posttreatment (Triantafyllou et al, 2018).

We do not know the mechanism that underlies the extreme sensitivity of $Axl^{-/-}$ mice to APAP, but one possibility is revealed by our finding that mRNA levels for matrix metalloprotease (MMP) 12 are markedly lower, specifically in these mutants (Fig 6F). This is of interest given that a recent study documented increases in hemorrhage and liver injury in $MMP12^{-/-}$ mice after APAP treatment that are very similar to the damage we observe in $Axl^{-/-}$ mice after APAP (Kopec et al, 2017). It is therefore possible that reduced expression of MMP12 in $Axl^{-/-}$ mice may be a significant contributor to the severe APAP phenotype that we observe in these animals.

### Role of TAM receptors in response to lethal liver injury

We also observed TAM regulation of the response to a lethal liver injury—a high (1 mg/kg) dose of Jo2—but with differences in the relative requirement for Axl and Mer. This dose led to AC accumulation at 2 and 4 h both in WT and in $Axl^{-/-}Mertk^{-/-}$ mice, although tissue damage at 2 h was much worse in the latter (Fig 7A). It resulted in the rapid activation of both Mer and Axl in the WT liver (Fig 7B), although Axl activation was obscured by metalloprotease cleavage of the Axl ectodomain and consequent reduction of full-length Axl protein in the liver after Axl activation (Zagórska et al, 2014) (Fig 7B). In contrast, the levels of full-length Mer were only modestly lowered by Jo2 (Fig 7B). Axl cleavage resulted in the appearance of elevated soluble Axl extracellular domain (sAxl) in the blood at 2 h after Jo2 injection (Fig 7C), suggesting that circulating sAxl could serve as a biomarker of acute liver damage. Consistent with these sAxl observations, we detected a dramatic loss in both steady-state Axl and Gas6 (which is normally complexed with Axl [Zagórska et al, 2014]) in the liver by immunohistochemistry at 2 h after a lethal Jo2 dose, with only a minimal reduction in the expression of Mer (Fig S2A), in keeping with the result seen by Western blot (Fig 7B). Jo2 lethality was significantly enhanced in $Axl^{-/-}Mertk^{-/-}$ mice relative to WT, but also in $Mertk^{-/-}$ and especially $Axl^{-/-}$ single mutants (Fig 7D). This may be related to TAM suppression of stimulus-induced inflammatory cytokine production in dendritic cells and macrophages (Sharif et al, 2006; Rothlin et al, 2007; Lemke & Rothlin, 2008) because we observed higher levels of TNFα (Tnf), type I interferon (Ifn), and IL-1β (Il1b) mRNAs in $Axl^{-/-}Mertk^{-/-}$ liver relative to WT after Jo2 administration (Fig 7E). Notably, Ifna4 and Ifnb mRNAs were up-regulated

±treatment with 10 nM Gas6, were immunoprecipitated and blotted for Axl. The low residual level of Axl immunostaining in KCs of these mice (A) may be due to the presence of a low level of a truncated Axl (asterisk) produced by Cre-mediated recombination. **(D)** Levels of the indicated mRNAs, isolated from 6 to 7 mo livers of the indicated genotypes, and quantified by qRT-PCR. $t$ test. **(E)** Cleaved Casp3$^+$ apoptotic cell accumulation in the liver 24 h after a nonlethal Jo2 injection in Tx-injected $Cx3cr1^{CreER/+}Axl^{f/f}Mertk^{f/f}$ mice, in which Axl and Mer expression in KCs is maintained (left), versus $Cx3cr1^{Cre/+}Axl^{f/f}Mertk^{f/f}$ mice, in which Axl and Mer expression in KCs is lost (right). Bars (A, B, E): 100 $\mu$m.

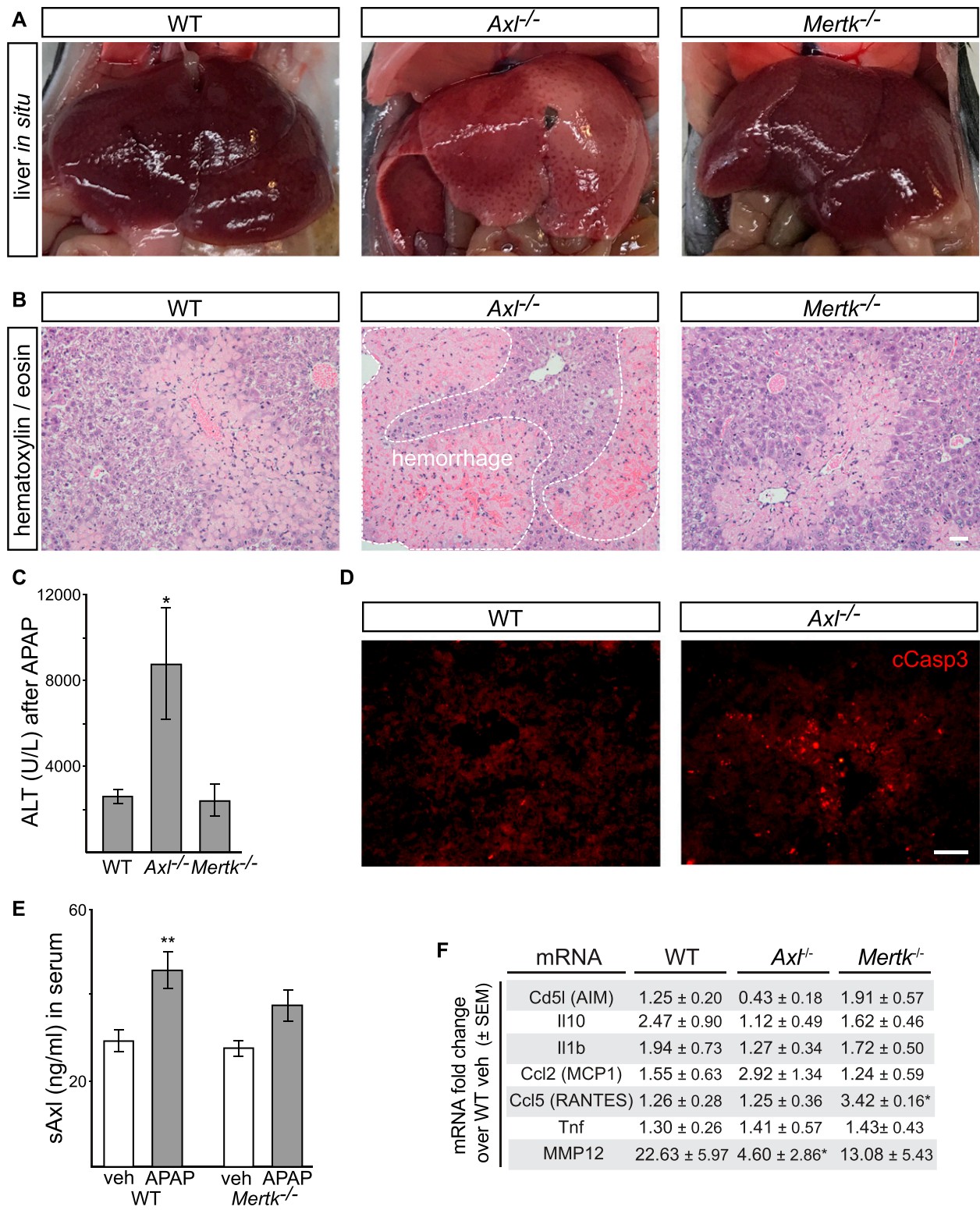

**Figure 6. Protective role of Axl in APAP intoxication.**
**(A)** Representative images showing extensive hemorrhage and congestion of liver lobes 48 h after administration of acetaminophen (APAP; 300 mg/kg) to $Axl^{-/-}$ but not $WT$ or $Mertk^{-/-}$ mice. These in situ liver images are from *non-perfused* mice. **(B)** Representative liver sections from 48 h APAP-treated mice were analyzed by H&E staining. Markedly increased congestion and blood hemorrhage is observed in $Axl^{-/-}$ but not $WT$ or $Mertk^{-/-}$ liver. Bar: 100 $\mu$m. **(C)** Measurement of circulating ALT (U/l, units per liter) 48 h after APAP administration in mice of the indicated genotypes. **(D)** Immunostaining for cleaved Casp3[+] cells in liver sections of WT and $Axl^{-/-}$ mice at 48 h after APAP administration. Bar: 50 $\mu$m. **(E)** Induction of soluble Axl (sAxl) in WT and $Mertk^{-/-}$ mice 48 h after APAP. The trend toward sAxl induction in $Mertk^{-/-}$ mice is not statistically

specifically in the $Axl^{-/-}$ and $Axl^{-/-}Mertk^{-/-}$ but not the $Mertk^{-/-}$ liver, whereas TNFα was up-regulated specifically in the $Mertk^{-/-}$ and $Axl^{-/-}Mertk^{-/-}$ but not the $Axl^{-/-}$ liver (Fig S2B–K).

### Role of TAM receptors in liver fibrosis

Finally, we examined TAM signaling during carbon tetrachloride ($CCl_4$) toxicity—a model of chronic liver damage (Weber et al, 2003) in which WT, $Axl^{-/-}$, $Mertk^{-/-}$, and $Axl^{-/-}Mertk^{-/-}$ mutants were injected with $CCl_4$ three times per week for 6 wk. We observed strikingly different results from those seen in the acute injury models: $CCl_4$-driven hepatic fibrosis was specifically *enhanced* by Axl signaling (Fig 8). Collagen deposition was comparable in WT, $Mertk^{-/-}$, and $Axl^{-/-}Mertk^{-/-}$ livers but was *reduced* in the $Axl^{-/-}$ liver (Fig 8A and D). Correspondingly, deposition of collagen-associated laminin was also reduced only in the $Axl^{-/-}$ liver (Fig 8B and E). In contrast, the phagocytic clearance of ACs was, as seen for the acute injury models, again entirely dependent on Mer (Fig 8C and F), which may account for the lack of fibrosis protection seen in the $Axl^{-/-}Mertk^{-/-}$ liver. Long-term $CCl_4$ exposure led to a substantial up-regulation of hepatic Axl and Gas6 (Fig 8G and H). Axl up-regulation was modestly associated with KCs and activated stellate cells (Fig S3A–C), but most prominently with loci of infiltrating monocytes (Fig S3D), which play critical roles in the progression of both acute and chronic liver disease (Tacke, 2017). Our conclusions from the $CCl_4$ fibrosis models are in substantial agreement with previously published analyses (Barcena et al, 2015).

## Discussion

Together, our results demonstrate that TAM RTKs are critical regulators of hepatic physiology and homeostasis. The livers of normally aged $Axl^{-/-}Mertk^{-/-}$ mice display substantially elevated AC accumulation, pronounced immune activation, and marked hepatic tissue damage relative to their WT counterparts, in the absence of any experimental perturbation. These findings indicate that sustained Axl and Mer signaling throughout adult life is required for healthy aging of the liver.

Significantly, Mer alone is required for the phagocytosis of ACs that are generated in settings of acute liver damage induced by Jo2, and Mer and Axl act in concert to suppress inflammatory responses in the liver, as they do in other organs. These findings, together with our observation that KCs are a particularly important locus of Mer action, are consistent with the fact that relatively high steady-state expression of Mer is a defining feature of phagocytic tissue macrophages throughout the body (Gautier et al, 2012; Lemke, 2019) and with the demonstration that Mer is absolutely required for the efficient clearance of ACs by these cells (Scott et al, 2001; Lemke & Burstyn-Cohen, 2010; Zagórska et al, 2014; Fourgeaud et al, 2016; Lemke, 2017).

In striking contrast to the Jo2 acute injury model, the most dramatic effects of TAM deletion in the APAP intoxication model are not seen with $Mertk^{-/-}$ mice. Whereas these $Mertk$ mutants display a ~1.4-fold increase in necrotic cells and an approximately twofold increase in activated neutrophils 24 h after APAP administration (Triantafyllou et al, 2018), many $Axl^{-/-}$ mice were at or near death from hepatic congestion and hemorrhage by 48 h after the same APAP treatment. It is important to note that these phenotypes—sinusoidal congestion and hemorrhage—are common features of acetaminophen-induced liver injury. Recently, it was revealed that MMP12 levels are increased in the liver after APAP overdose, and that targeted deletion of MMP12 increases sinusoidal congestion and hemorrhage and hepatocellular necrosis after APAP treatment (Kopec et al, 2017). Although the mechanism by which MMP12 functions to maintain sinusoidal integrity remains unknown, our studies suggest that Axl is a critical regulator of MMP12. Accordingly, Axl activation on sinusoidal ECs or KCs may limit sinusoidal destruction by increasing levels of MMP12. Although it is not known whether Axl directly regulates MMP12, studies have shown that Axl does regulate other MMPs, including MMP1, 2, 3, and 9 (Xu et al, 2013; Divine et al, 2016).

The co-expression of Mer and Axl together in KCs, which is also seen in red pulp macrophages and select other macrophage populations but not in tissue macrophages generally, may account for the concerted immunosuppressive action of these two receptors in the liver. Immunosuppression by KCs is critical because the liver is continuously exposed to endotoxin and other microbial products present in the portal circulation (Seo & Shah, 2012). In the absence of TAM-driven immunosuppression, liver inflammation and tissue damage progress rapidly with age.

Soluble Axl ectodomain (sAxl) has been reported to be an accurate biomarker of cirrhosis and the development of hepatocellular carcinoma (Dengler et al, 2017). Our results provide a mechanistic explanation for sAxl release subsequent to Axl activation in the presence of ACs. In settings of cirrhosis, this activation may occur not only in KCs, but also in infiltrating monocytes, activated stellate cells, and even vascular ECs.

In contrast to the beneficial TAM effects seen in acute injury, Axl signaling is deleterious during chronic hepatic fibrosis, as it promotes scarring. This latter finding is consistent with earlier work demonstrating improvement of $CCl_4$-induced steatohepatitis and fibrosis in $Gas6^{-/-}$ and $Axl^{-/-}$ mice (Fourcot et al, 2011; Barcena et al, 2015). It has previously been reported that hepatic stellate cells elevate Axl expression upon activation (Barcena et al, 2015; Zhang et al, 2016), and studies have indicated that AC phagocytosis by HSCs activates them and stimulates collagen production (Zhan et al, 2006; Jiang et al, 2009). Our results suggest that Axl expression by KCs, activated stellate cells, and infiltrating monocytes together contribute to the deleterious role of Axl in hepatic fibrosis, although the relative contribution of each cell type remains to be determined.

Many different TAM-targeting agents, including biologics and small molecule kinase inhibitors, are currently in preclinical

---

significant. As for all APAP treatments, mice were fasted for 16 h before drug administration. **(F)** Levels of the indicated mRNAs, isolated livers of the indicated genotypes 48 h after APAP treatment, and quantified by qRT-PCR. *$P < 0.05$; **$P < 0.005$. Two-way ANOVA (Bonferroni multiple comparison correction) (C, E); $t$ test (F).

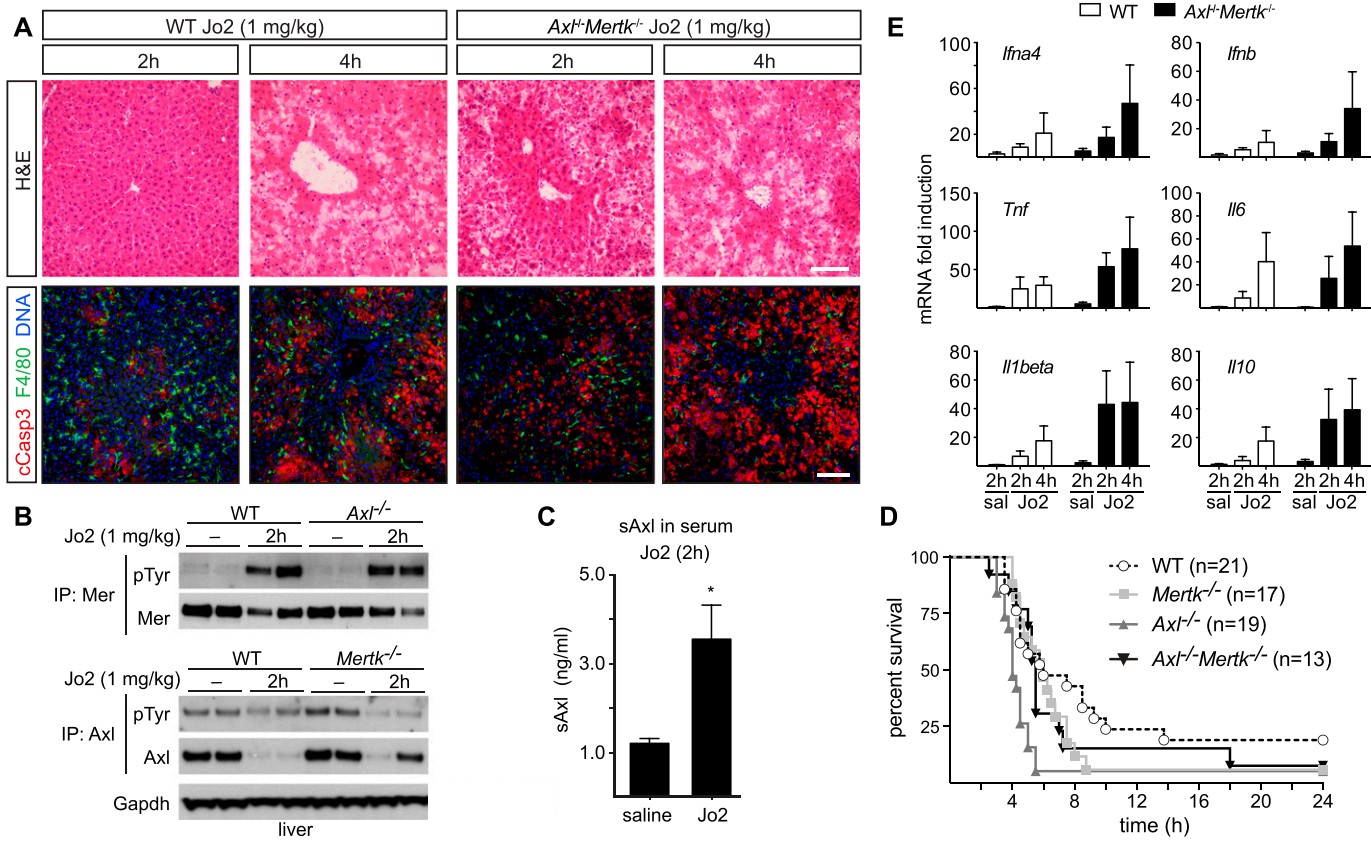

**Figure 7. Protective role of Axl and Mer in lethal Jo2 liver damage model.**
**(A)** Liver sections at 2 and 4 h after a lethal i.p. injection (1 mg/kg) of Jo2 were analyzed by H&E staining and immunohistochemistry with indicated antibodies. Increased accumulation of apoptotic cells can be observed in $Axl^{-/-}Mertk^{-/-}$. Bars: 100 $\mu$m. Representative images from three experiments (n = 2–5 mice per genotype). **(B)** Liver lysates from WT and $Axl^{-/-}$ mice injected with saline or Jo2 (1 mg/kg) were immunoprecipitated for Mer and Axl and immunoblotted with indicated antibodies; n = 2, each lane an individual mouse. **(C)** sAxl ELISA in serum of saline and Jo2 (1 mg/kg) injected mice; n = 2, *P < 0.05. t test. **(D)** Survival of mice of the indicated genotypes after Jo2 (1 mg/kg) injection. **(E)** Expression of the indicated inflammatory markers was analyzed by qRT-PCR from liver mRNA samples; representative results from one to three experiments.

development for the treatment of multiple types of cancers, and several have entered clinical trials. Our results have obvious and immediate implications for the potential re-purposing of these agents to the treatment of liver disease, as they demonstrate divergent activities for Mer and Axl in distinct settings of acute and chronic liver injury. The most salient of these relates to specificity: Axl inhibitors used to treat fibrosis must not perturb the Mer activity required for the response to acute liver injury, and correspondingly, Mer agonists used to treat acute injury must not stimulate the Axl activity that promotes fibrosis.

# Materials and Methods

## Mice

The $Axl^{-/-}$, $Mertk^{-/-}$, $Axl^{-/-}Mertk^{-/-}$, $Cx3cr1^{GFP/+}Axl^{-/-}Mertk^{-/-}$, and $Cx3cr1^{Cre/+}Axl^{f/f}Mertk^{f/f}$, $Cx3cr1^{CreER/+}Axl^{f/f}Mertk^{f/f}$ mice were as described previously (Lu et al, 1999; Yona et al, 2013; Fourgeaud et al, 2016; Schmid et al, 2016). All lines have been backcrossed for >9 generations to a C57BL/6 background. All animal procedures were conducted according to guidelines established by the Salk

Institutional Animal Care and Use Committee. Mice were randomly allocated to experimental groups. Mice were fed irradiated rodent diet 15053 (Lab Diet), caged in individual ventilated cages with Anderson 0.25-inch corn cob bedding, and maintained on a 12-h light–dark cycle.

## Acute liver damage models

In the fulminant hepatitis (Jo2) model, mice were injected with an anti-CD95 monoclonal antibody (Jo2 clone, Cat. no. 554254; BD Pharmingen), which activates the Fas receptor and induces apoptosis of hepatocytes (Ogasawara et al, 1993; Yin et al, 1999). For a lethal dose, 8–12-wk-old mice (males and females) were injected i.p. with 1 mg/kg body weight of antimouse CD95 antibody (Jo2) and monitored every 30 min. For a sublethal dose, 8–12-wk-old mice were injected i.p. with 0.3 mg/kg Jo2 antibody. The D-galactosamine (D-gal)/LPS model mimics bacterial peritonitis and endotoxic shock (Lehmann et al, 1987). For a sub-lethal dose, 8–12-wk-old mice (males and females) were injected i.p. with D-gal (Cat. no. G0500, 350 mg/kg body weight; Sigma-Aldrich) and LPS (Cat. no. ALX-581-013-L002, 1 $\mu$g/kg body weight; ENZO Life Sciences) in saline (final volume 0.3–0.4 ml). Mice were euthanized at the indicated times

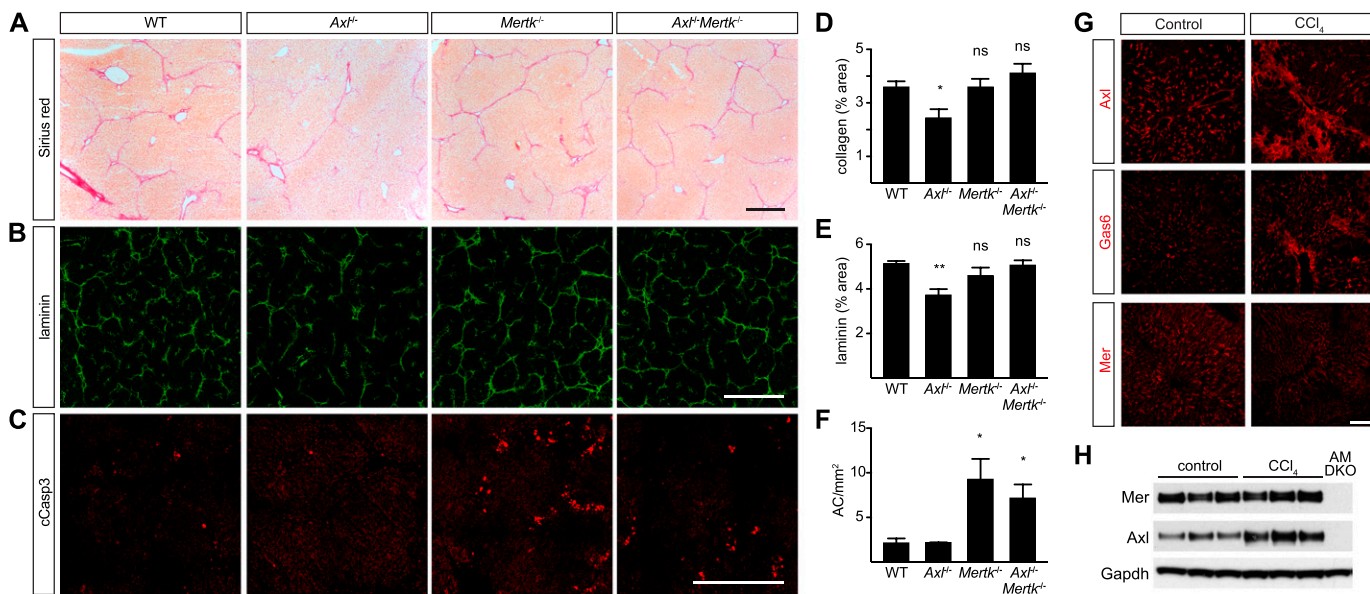

**Figure 8. Axl promotion of CCl₄-induced liver fibrosis.**
**(A)** Sirius red staining, showing collagen deposition, of liver sections from mice of the indicated genotypes, injected with CCl₄ three times per week for 6 wk. **(B)** Laminin staining of similar sections. **(C)** Cleaved caspase 3 (cCasp3) staining of similar sections. **(A, B, C, D, E, F)** Quantification of the results in (A, B, C), respectively. **(G)** Up-regulation of Axl and Gas6, but not Mer, in fibrotic liver. **(H)** Western blot showing increased expression of Axl in fibrotic liver. **(A, B, C, G)** Bars (A, B, C): 0.5 mm (G) and 100 μm. Representative images from six mice per genotype. *P < 0.05, **P < 0.01. t test (D, E, F).

postinjection. Serum and liver samples were collected for histological and biochemical analysis.

For the acetaminophen toxicity model, mice were treated with 300 mg/kg acetaminophen (10 μl/g) (Sigma-Aldrich Chemical Company) in sterile saline, or sterile saline vehicle alone, by i.p. injection. Mice were fasted in clean cages for 16 h (overnight) before injection the following morning. (Fasting is necessary to produce acetaminophen-induced liver injury in mice.) Mice were monitored for general appearance, motility, and survival at 12, 24, and 48 h after treatment. Although the goal of these studies was to evaluate the impact of TAM deletion on the resolution of inflammation, which occurs between 48 and 72 h after acetaminophen overdose (Zigmond et al, 2014), this was precluded by the severe phenotypes that appeared in the Axl⁻/⁻ mice as soon as 12 h after treatment. Liver and blood were collected from mice at 48 h after acetaminophen. At 48 h, mice were anesthetized with pentobarbital (50 mg/kg) and euthanized by exsanguination.

### Carbon tetrachloride liver fibrosis model

Carbon tetrachloride (CCl4; Sigma-Aldrich) was injected i.p. at 0.5 ml/kg, diluted in corn oil (Sigma-Aldrich) at 0.2 ml final volume, three times per week for 6 wk (Ding et al, 2013). 2 d after the last injection, mice were euthanized and serum and liver were collected for histological and biochemical analysis.

### Antibodies

Antibodies used were Mer for WB (AF591; R&D Systems), Mer for IHC (DS5MMER; eBioscience), Axl for IHC (AF854; R&D Systems), Axl for immunoprecipitation (M-20; Santa Cruz), GAPDH (MAB374, clone 6C5;

Millipore), phospho-tyrosine (clone 4G10; Millipore), Gas6 (AF986; R&D Systems), CD31 (ab28364; Abcam), MARCO (MCA1849; AbD Serotec), Laminin (L9393; Sigma-Aldrich), Desmin (ab32362; Abcam), cleaved Caspase 3 (Asp175; Cell Signaling), F4/80 (MCA497; AbD Serotec), and Ki67 (65241; BioLegend). Axl, Mer, and Gas6 antibody specificity for immunohistochemistry, immunocytochemistry, immunoprecipitation, and immunoblotting was tested using samples from corresponding mouse mutants (e.g., Fig 1G and H). Secondary antibodies used for immunoblot analysis were horseradish-peroxidase–conjugated anti-goat (705-035-003) from Jackson ImmunoResearch and antimouse (NA931V) and antirabbit (NA934V) from GE Healthcare. Secondary antibodies for immunocyto- and immunohistochemistry were fluorophore-conjugated antigoat (A-11055 from Life Technologies, or 705-166-147 from Jackson ImmunoResearch), antirabbit (A-10040 or A-21206 from Life Technologies), antirat (712-545-153 or 712-165-153 from Jackson ImmunoResearch), and antimouse (A-11029 from Life Technologies, or 715-166-150 from Jackson ImmunoResearch).

### KC isolation

Non-parenchymal cells (NPCs) were prepared by two-step in situ/ ex situ collagenase/pronase digestion and fractionation on a continuous density gradient of 36% Percoll prepared in GBSS/B (Gey's Balanced Salt Solution supplemented with 0.8% NaCl at pH 7.35). The procedure was carried out in anesthetized animals (100:10 mg/kg Ketamine:Xylazine). Perfusion was performed in situ via portal vein with a 24Gx3/4″ catheter. HBSS without Ca²⁺ was perfused at 37°C with a flow rate of 10 ml/min. During the first 5 min of perfusion, regular HBSS was used as a washing solution. Subsequently, HBSS containing 0.04% Collagenase Type 2 (Worthington),

50 mM Hepes (N-2-hydroxyethylpiperazine-N'-2-ethanesulfonic acid), and 0.6 mg/ml CaCl$_2$*2H$_2$O was substituted, and the perfusion continued for 5 min. The perfused liver was excised, minced ex situ, and mixed with 30 ml of HBSS containing 0.07% Pronase E (Merck EMD), 50 mM Hepes, and 0.6 mg/ml CaCl$_2$*2H$_2$O. 20 min after continuous stirring at 37°C, the cell suspension generated was filtered through a 100-$\mu$m cell strainer (Falcon), and the filtrate was centrifuged three times for 5 min (50$g$) in 50 ml of HBSS supplemented with 0.2 mg/ml EGTA and 20 $\mu$g/ml DNAse-I, to pellet the hepatocytes. The three supernatants obtained were used for the preparation of KCs by centrifugation over a 36% Percoll continuous density gradient (density, 1.066 g/ml; pH 7.5 in GBSS/B). Hepatocyte-depleted supernatants were subjected to a second centrifugation (600$g$, 4°C, 5 min), and the pellets were mixed, resuspended with 40 ml Percoll-GBSS/B Solution (36% Percoll), transferred in one 50-ml tube, adding 500 $\mu$l DNAse-I (2 mg/ml), and centrifuged in a swinging bucket rotor during 20 min (800$g$, 4°C) without brakes. To remove the Percoll, the pellet was resuspended in 14 ml of GBSS/B and centrifuged 5 min (800$g$, 4°C). The pellet was resuspended in 4 ml of Red Blood Lysis Buffer during 10 min (00-4333-5710; eBioscience) to remove erythrocyte contamination and centrifuged 5 min (800$g$, 4°C). The final precipitate is an enriched fraction containing NPCs, but it is practically free of debris and hepatocytes. KCs were selectively removed from the fractions by selective attachment to plastic. The resuspended NPC pellet was plated at 10 × 6 cells per cm$^2$ in six-well cluster dishes (Becton Dickinson) and cultured in RPMI 1640 supplemented with 10% FCS and antibiotics (50 $\mu$g/ml each of penicillin, streptomycin, and gentamicin). KCs were allowed to attach to the plastic for 30 min and were washed several times with PBS to remove unattached cells.

## Immunoblotting and immunoprecipitation

Thioglycolate-elicited peritoneal macrophages were washed with ice-cold Dulbecco's phosphate-buffered saline and lysed on ice in a buffer containing 50 mM Tris–HCl, pH 7.5, 1 mM EGTA, 1 mM EDTA, 1% Triton X-100, 0.27 M sucrose, 0.1% $\beta$-mercaptoethanol, and protease and phosphatase inhibitors (Roche). Tissues were snap-frozen in liquid nitrogen before lysis. For immunoblots, equal amounts of protein (10 $\mu$g) in lithium dodecyl sulfate sample buffer (Invitrogen) were subjected to electrophoresis on 4–12% Bis-Tris polyacrylamide gels (Novex; Life Technologies) and transferred to polyvinylidene difluoride membranes (Millipore). For immuno-precipitations, cell lysates were incubated overnight (ON) at 4°C with indicated antibodies. Protein G-Sepharose (Invitrogen) was added for 2 h and immunoprecipitates (IPs) were washed twice with 1 ml of lysis buffer containing 0.5 M NaCl and once with 1 ml of 50 mM Tris–HCl, pH 7.5. IPs were eluted in lithium dodecyl sulfate buffer, separated on polyacrylamide gels, and transferred to polyvinylidene difluoride membranes. Membranes were blocked in TBST (50 mM Tris–HCl, pH 7.5, 0.15 M NaCl, and 0.25% Tween-20) containing 5% BSA and immunoblotted ON at 4°C with primary antibodies diluted 1,000-fold in blocking buffer. The blots were then washed in TBST and incubated for 1 h at 22–24°C with secondary HRP-conjugated antibodies (GE Healthcare) diluted 5,000-fold in 5% skim milk in TBST. After repeating the washes, signal was detected with enhanced chemiluminescence reagent. Image quantification was performed using ImageJ software.

## sAxl, TNF $\alpha$, and IL-1 $\beta$ ELISA assays

ELISA for measurement of sAxl, Gas6 (R&D Systems), TNF $\alpha$, and IL-1 $\beta$ (eBiosciences) was performed according to the manufacturers' instructions.

## ALT and AST assay

ALT (TR71121) and AST (TR70121) assays were from Thermo Fisher Scientific and were performed according to the manufacturer's instruction.

## Sirius red stain

Sections were fixed for 24 h in PFA/BSA 4% at RT, washed in PBS two times for 5 min, and stained with hematoxylin for 8 min at RT. Slides were then washed twice in lukewarm tap water for 5 min and stained for 1 h at RT with Picro-Sirius Red solution (0.1% [m/v] Sirius Red [Sigma-Aldrich] in saturated aqueous solution of picric acid [LabChem]). Slides were washed two times for 5 min with "acidified water" (acetic acid 0.5%) and dehydrated in 100% EtOH three times for 2 min. Finally, the slides were cleared in Histo-Clear (National Diagnostics) for 5 min and mounted with VectaMount (Vector Labs).

## Immunocytochemistry and immunohistochemistry

For immunohistochemistry, tissues were fresh-frozen and cut into 11-$\mu$m sections, air-dried, and stored desiccated at −70°C. Before staining, sections were fixed for 3 min with ice-cold acetone, washed in PBS, blocked in blocking buffer (PBS containing 0.1% Tween-20, 5% donkey serum and 2% IgG-free BSA) for 1 h. Slides were then washed in PBS 0.1% Tween-20 and incubated with 1 $\mu$g/ml primary antibody in blocking buffer ON at 4°C. The slides were washed five times for 5 min in PBS 0.1% Tween-20 and incubated with Hoechst and fluorophore-coupled donkey (Jackson) secondary antibodies diluted 1:400 in blocking buffer for 2 h at 22–24°C in dark. Slides were washed, sealed with Fluoromount-G (SouthernBiotech), and stored at 4°C. For immunocytochemistry, cells were first fixed in 4% PFA/BSA for 10 min and washed with PBS. Cover slips were then blocked for 30 min in blocking buffer with 0.1% Triton X-100, washed in PBS 0.1% Tween-20, and incubated with primary antibody for 1 h at 22–24°C. Cover slips were washed five times in PBS 0.1% Tween-20 and incubated with Hoechst and fluorophore-coupled donkey secondary antibody diluted 1:400 in blocking buffer for 1 h at 22–24°C in dark. Cover slips were washed and mounted on slides with Fluoromount-G. Images were taken on a Zeiss LSM 710 microscope with Plan-Apochromat 20×/0.8 M27 and 63×/1.40 Oil DIC M27 objectives. Morphometric quantification was performed using Image J to quantify % marker area or number of positive cells per area. 10–15 images from three separate sections per mouse were quantified.

## qRT–PCR

Total cellular RNA was isolated using TRIzol reagent (Thermo Fisher Scientific), according to manufacturer's instructions. Reverse transcription was performed with RT Transcriptor First Strand cDNA Synthesis Kit (Roche) with anchored oligo-dT primers (Roche). qRT-PCR was run in a 384-well plate format on a ViiA 7 Real-Time PCR System (Applied Biosystems) using 2× SYBR Green PCR Master Mix (Applied Biosystems). Analysis was performed using delta-delta Ct method. Primers are listed in Table S1. Cyclophilin A (Ppia) and 36B4 (Rplp0) were used as control housekeeping genes.

## Data analysis

Statistical analysis was performed using two-tailed *t* test or ANOVA (Dunnett's or Bonferroni multiplicity correction was used for multiple comparison adjustments).

## Supplementary Information

## Acknowledgements

We thank Joe Hash for excellent technical assistance. This work was supported by grants from the NIH (R01 NS085296, R01 AI101400, and P30CA014195), the Leona M and Harry B Helmsley Charitable Trust (#2012-PG-MED002), the Nomis, HN and Frances C Berger, Fritz B Burns, and HKT Foundations (to G Lemke) and by postdoctoral fellowships from the Marie Curie International Outgoing Fellowship Program (to PG Través), the Nomis Foundation (to A Zagórska), and the Fundación Alfonso Martín Escudero (to L Jiménez-García).

## Author Contributions

A Zagórska: conceptualization, resources, data curation, software, formal analysis, supervision, validation, investigation, visualization, methodology, and writing—review and editing.
PG Través: conceptualization, resources, data curation, formal analysis, supervision, validation, investigation, methodology, and writing—review and editing.
L Jiménez-García: conceptualization, data curation, formal analysis, investigation, methodology, and writing—review and editing.
JD Strickland: data curation, formal analysis, investigation, and methodology.
J Oh: data curation, formal analysis, and investigation.
FJ Tapia: data curation and visualization.
R Mayoral: conceptualization, data curation, and formal analysis.
P Burrola: data curation, formal analysis, and visualization.
BL Copple: conceptualization, data curation, formal analysis, investigation, methodology, and writing—review and editing.
G Lemke: conceptualization, data curation, formal analysis, supervision, funding acquisition, project administration, and writing—original draft, review, and editing.

## Conflict of Interest Statement

A Zagórska and PG Través are currently employees of Gilead Sciences, Inc., J Oh is currently an employee of Erasca, Inc., and R Mayoral is currently an employee of Merck & Co. All authors declare no conflicts of interest.

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
