## [Reviewer comments · Life Science Alliance]

Life Science Alliance

Differential regulation of hepatic physiology and injury by the TAM receptors Axl and Mer

Anna Zagórska, Paqui Través, Lidia Jiménez-García, Jenna Strickland, Joanne Oh, Francisco Tapia, Rafael Mayoral Monibas, Patrick Burrola, Bryan Copple, and Greg Lemke

DOI: <https://doi.org/10.26508/lsa.202000694>

Corresponding author(s): Greg Lemke, Salk Institute for Biological Studies

Review Timeline:	Submission Date:	2020-03-11
	Editorial Decision:	2020-04-07
	Revision Received:	2020-05-11
	Editorial Decision:	2020-05-25
	Revision Received:	2020-05-28
	Accepted:	2020-05-29

Transaction Report:

April 7, 2020

Re: Life Science Alliance manuscript #LSA-2020-00694

Dr. Greg E. Lemke
Salk Institute for Biological Studies
10010 North Torrey Pines Road
La Jolla, CA 92037

Dear Dr. Lemke,

Thank you for submitting your manuscript entitled "Differential TAM receptor regulation of hepatic physiology and injury" to Life Science Alliance. The manuscript was assessed by expert reviewers, whose comments are appended to this letter.

As you will see, the reviewers appreciate your data and are supportive of further consideration of your manuscript for publication here. However, they also raise some concerns that will need to get addressed. We would thus like to invite you to submit a revised version of your manuscript to us. Both reviewer #1 and #2 point out that single KO analyses should get added to emphasize the importance of the double KO in age-related liver damage. Reviewer #3 furthermore thinks that the double KO findings should get supported by additional insight into inflammation, necrosis, and fibrosis. Also, some clarifications and quantifications are needed.

We are aware that many laboratories cannot function fully during the current COVID-19/SARS-CoV-2 pandemic and therefore encourage you to take the time necessary to revise the manuscript to the extent requested. We will extend our 'scoping protection policy' to the full revision period required. You also may have data already at hand to address the major criticisms raised, and I would be happy to discuss with you how you could address the specific requests made. Maybe we can speak on the phone or via skype, or you can send me a point-by-point response outline via response email.

Thank you for this interesting contribution to Life Science Alliance. We are looking forward to receiving your revised manuscript.

Sincerely,

B. MANUSCRIPT ORGANIZATION AND FORMATTING:

Reviewer #1 (Comments to the Authors (Required)):

Zagorska and Traves et al. describe in their submitted manuscript entitled "Differential TAM receptor regulation of hepatic physiology and injury" the involvement of Mer and Axl in liver homeostasis during healthy aging, acute liver injury and liver fibrosis. In a translational approach using mouse models of acute liver injury (Jo2, LPS/ D-galactosamine and acetaminophen) and a liver fibrosis model (CCl4), authors show the pro-inflammatory role of Mer and Axl and the and pro-fibrotic role of Axl.

These results are of potential interest, showing the relevance of Axl and Mer in response to acute liver injury and fibrosis. The study is well structured and the manuscript is well presented in its current form. The overall merit of the study supports to be accepted for publication with some revisions.

Issues:

1. Title could be more specific by focusing on Axl and Mertk
2. Studies on the role of TAM receptors in hepatic aging should also be done with single Axl $-/-$ and Mertk $-/-$ mice.
3. Fig 3B: Authors describe an accumulation of apoptotic and necrotic cells in the liver. How do they estimate necrosis since apoptosis is determined by the number of cleaved Caspase 3+ cells?
4. Fig 3F: Authors should quantify results of IP to clarify if there is no increase in Axl phosphorylation.
5. Fig 3 / LPS / D-galactosamine model: The authors should show increased Mertk activation via IP as shown for the Jo2 model.
6. Fig 5A-C: It is unclear why authors use the tamoxifen inducible system.
7. Fig 6 / Acetaminophen model: No explanation / hypothesis is provided by the authors why Axl $-/-$ mice react so harsh and how Axl signaling is protective in this context. Authors should show increased Axl activation via IP as shown for Mertk in the Jo2 model (Fig 3).
8. Fig 6D: indication of staining missing (cleaved Casp3)
9. sAxl levels in Mertk $-/-$ mice should be included in Fig. 6E (and Fig 7C).
10. Fig. 7F: Is there a statistical significance between wt and Axl $-/-$ /Mertk $-/-$?
11. Fig. 8: CCl4 induced fibrosis: Axl identified as a driver of fibrosis as Axl $-/-$ mice show reduced fibrosis. Why is no reduction of fibrosis detected in Axl $-/-$ /Mertk $-/-$ mice?
12. Suppl Fig S2 is misleading. What are the differences between the two upper rows in panel A and C, both showing Axl expression in the control?
13. It should be shown whether Axl expression is induced in hepatocytes upon CCl4 treatment.
14. Authors should discuss data of human Axl and Mertk expression from patients with liver fibrosis.

Reviewer #2 (Comments to the Authors (Required)):

Zagórska, Través et al is a very interesting addition to our knowledge on the roles of AXL and MERTK in different pathological conditions in the liver. The manuscript provides a large amount of consistent data, revealing the specific role of each receptor.

Expression of TAM receptors in mouse liver: Strong expression of both Axl and Mer in Kupffer cells, Axl and Mer also expressed in many CD31+ endothelial cells of the liver vasculature, with Axl in strongly CD31+ blood vessels and Mer in weakly CD31+ hepatic sinuses. Axl was also expressed in perivascular macrophages. The histological characterization of the system is very conclusive.

Role of TAM receptors in the aging process of the liver: Aged Axl $-/-$ Mertk $-/-$ livers were damaged and inflamed in the absence of any experimental insult, suggesting TAM signaling is required for normal liver homeostasis and healthy aging. The data are supportive, however, if the authors could

provide information from single AXL^{-/-} and Mertk^{-/-} aged mice it would emphasize the importance of the double suppression in age-related liver damage.

Role of TAM receptors in Jo2 and LPS/D-Gal acute injury models: In a fulminant hepatitis model based on injection of the Jo2 anti-Fas antibody only Mertk^{-/-} and Axl^{-/-} Mertk^{-/-} mice presented liver damaged and increased cleaved caspase-3. Minor point: Caspase 3 activation in the case of Axl^{-/-} animals (Fig 3B) seems intermediate compared to WT and MerTK mice. Is this correct?

Similarly, in an endotoxic shock model after LPS and D-galactosamine injection, liver damaged and apoptotic cells were detected at 16 h only in Mertk^{-/-} and Axl^{-/-} Mertk^{-/-} mice, also exhibiting less recovery and regeneration after 7 days. The data are very convincing. The western showing AXL phosphorylation may suggest some increase, could you provide the quantification of AXL and MERTK p-Tyr in these experiments? In addition, have you checked for soluble AXL and MERTK serum levels after treatments?

TAM receptor expression in KCs is critical for liver physiology: Aged Cx3cr1Cre/+Axl^{fl/fl}Mertk^{fl/fl} livers displayed elevated levels of many proinflammatory and immunoregulatory mRNAs relative to Axl^{fl/fl}Mertk^{fl/fl}. The data are compelling.

Role of TAM receptors in acetaminophen-induced acute liver injury: The livers of APAP-treated mice at 48 hours after drug administration revealed massive liver damage specifically in the Axl^{-/-} mice. The data are supportive. Do you have any insight of the mechanism involved in Axl activation by APAP? Could this be a redox-mediated mechanism?

Role of TAM receptors in response to lethal liver injury: A high (1 mg/kg) dose of Jo2 induced AXL and MERTK phosphorylation, increased sAXL levels. AXL^{-/-} and Mertk^{-/-} mice were more sensitive to Jo2 challenge relative to WT, as well as Tyro3^{-/-} Axl^{-/-} Mertk^{-/-} mice. The data are clear. Are soluble levels of MERTK in serum increased after Jo2 administration? Is there a reason why TKO are shown whereas Axl^{-/-} Mertk^{-/-} are not?

Role of TAM receptors in liver fibrosis: CCl₄-driven hepatic fibrosis was specifically enhanced by Axl signaling and AXL^{-/-} mice protected. The data are strongly supportive.

Reviewer #3 (Comments to the Authors (Required)):

The paper by Zagorska et al. summarises important data, which for the first time demonstrate the importance of constitutive TAM receptor Axl and Mertk expression on KCs for the maintenance of liver tissue homeostasis in the mouse. Their findings and conclusions are developed using a thorough methodology and convincing IHC/IF images, and confirmatory techniques. TAM receptor expression on macrophages and their role in liver tissue homeostasis are investigated under physiological conditions and mouse models of acute and chronic liver injury. Some models of acute liver injury highlight a pivotal role for Mertk in preventing acute liver injury (Jo2 and LPS/D-Gal), while another model (APAP) highlights the importance of Axl. A model of chronic liver injury (CCl₄ for 6 weeks) was used to evaluate the role of TAM receptor expression in fibrogenesis and suggests an association with AXL signalling.

Main points

1. The constitutive expression of Axl and Mer in KCs in adult mouse liver under physiological conditions was heretofore unknown and was clearly demonstrated by IHC (Fig 1). On page 11 the authors write "Most tissue macrophages express high and low levels of Mertk and Axl, respectively, ...". Please clarify or rephrase this statement. The image (Fig 1A, B and H) do not demonstrate the finding that Mertk is expressed at high and Axl at low levels on KC.
2. The authors conclude that aged livers from Axl^{-/-}Mertk^{-/-} animals were "damaged and inflamed" (page 12) having shown an up-regulation of markers for apoptosis, macrophage activation and

infiltration and other indirect markers. The findings should be supported by an H&E stain highlighting inflammation and possibly necrosis and fibrosis.

3. The development of acute liver tissue injury in experimental models in TAM knockout mice is convincingly demonstrated and very interesting. Fig 3B: in H&E the liver integrity looks better in Axl^{-/-}Mertk^{-/-} animals compared to Mertk^{-/-} animals. Is this just a matter of the picture chosen for the figure? In case this was a finding seen throughout the liver of all mice, are there any other pathways that may be compensatory up-regulated in Axl^{-/-}Mertk^{-/-} animals trying to maintain liver tissue homeostasis? Please review and append and/or comment. It is also interesting to see the recovery at day 7 - please add data for the Jo-2 model and include H&E in Fig 4F.

4. The conditional knockout mouse data supports the notion that TAM receptor expression in KC rather than LSEC is indeed controlling liver tissue homeostasis. It remains unclear why the role of Axl expression on peritoneal macrophages is stressed here and not mentioned in other sections of the manuscript.

5. The data highlighting a role for Axl in tissue protection against APAP induced acute liver injury are interesting. It would be also interesting to see the extend of APAP induced liver injury in the Axl^{-/-}Mertk^{-/-} animals model. Moreover in respect to the controversy to the data reported in Ref 48, the expression of Axl and Mertk in liver tissue in the different strains should be shown.

6. It is most interesting to find that different models of acute liver injury seem to induce different pathways and involve different homeostasis mechanisms in KCs. The data and potential mechanisms behind this should be discussed in more detail or further explored. This is highly relevant in respect to potential future studies in humans and allows no conclusion for therapeutic translation to human diseases at this point.

7. In Fig 7A expression levels of Axl and Mertk should be shown. Regarding soluble Axl (Fig 7C): Also soluble Mertk should be included here. It should further be discussed, that their source does not consist of KCs only - but there are several other TAM receptor expressing cells in the body that may contribute to the soluble fraction including circulating monocytes.

8. The expression of TAM receptors in the CCl₄ model shown here is new and interesting but studied in less detail compared to the acute liver injury models. Data showing inflammation and KCs in different strains following CCl₄ treatment are not shown. The bright Axl signal on IHC shown in Fig S2 are less convincing, the role of the EC is not addressed here, although part of the signal may be related to cells other than macrophages and HSC.

Replies to our Reviewers

Given current circumstances, we very much appreciate the time that our three referees have devoted to considering our manuscript, and are encouraged by their consistently positive evaluations of the work we describe. Outlined below are the additions and changes we have made in the revised paper to address the points they have raised.

Reviewer #1

These results are of potential interest, showing the relevance of Axl and Mer in response to acute liver injury and fibrosis. The study is well structured and the manuscript is well presented in its current form. The overall merit of the study supports to be accepted for publication with some revisions.

We thank Reviewer 1 for this generous assessment of our work.

Issues:

1. Title could be more specific by focusing on Axl and Mertk

We have changed the title to read: "Differential regulation of hepatic physiology and injury by the TAM receptors Axl and Mer".

2. Studies on the role of TAM receptors in hepatic aging should also be done with single Axl -/- and Mertk -/- mice.

We have now included additional (new) qPCR data on the levels of multiple cytokine and chemokine mRNAs in both *Axl*^{-/-} and *Mertk*^{-/-} single mutant mice at 6-8 months of age, in Supporting Fig. 1B-F.

3. Fig 3B: Authors describe an accumulation of apoptotic and necrotic cells in the liver. How do they estimate necrosis since apoptosis is determined by the number of cleaved Caspase 3+ cells?

Reviewer 1 is correct: we have only examined cCasp3 as a marker of apoptotic cells. We have removed the reference to necrotic cells from the Figure 3 legend.

4. Fig 3F: Authors should quantify results of IP to clarify if there is no increase in Axl phosphorylation.

We have now quantified the results displayed in Fig. 3F, and present this quantification in a new panel 3G. The non-lethal Jo2 dose does not result in Axl activation (auto-phosphorylation).

5. Fig 3 / LPS / D-galactosamine model: The authors should show increased Mertk activation via IP as shown for the Jo2 model.

We did not do this analysis.

6. Fig 5A-C: It is unclear why authors use the tamoxifen inducible system.

As outlined in the paper, these analyses allow us to demonstrate that elimination of TAM signaling specifically in mature Kupffer cells is critical for the phenotypes that develop during both aging and in the response to acute injury in the complete *Axl*^{-/-} *Mertk*^{-/-} double knock-outs.

7. *Fig 6 / Acetaminophen model: No explanation / hypothesis is provided by the authors why Axl^{-/-} mice react so harsh and how Axl signaling is protective in this context.*

We have now outlined an interesting hypothesis, based on the inclusion of new data. A recent study (Kopec *et al.*, *J. Hepatology*, 2017) documented increases in hemorrhage and liver injury in *MMP12^{-/-}* mice after APAP treatment that are very similar to the damage we observe in *Axl^{-/-}* mice after APAP. We have now found that *MMP12* mRNA levels are markedly lower in *Axl^{-/-}* mice relative to wild-type, and have included these new data in a revised panel F of Fig. 6. We hypothesize that reduced expression of *MMP12* in *Axl^{-/-}* mice may be a significant contributor to the severe phenotype that we observe in these mice subsequent to APAP treatment, and now mention this in both the Results and the Discussion.

8. *Fig 6D: indication of staining missing (cleaved Casp3)*

We have corrected this oversight, and added a label to Fig. 6D.

9. *sAxl levels in Mertk^{-/-} mice should be included in Fig. 6E (and Fig 7C).*

As requested, we have now included data on sAxl levels in the *Mertk^{-/-}* mice in response to APAP in Fig. 6E. The trend toward an increase is not statistically significant.

10. *Fig. 7F: Is there a statistical significance between wt and Axl^{-/-}/Mertk^{-/-}?*

Although the trends in these Fig. 7F analyses are apparent, the large variability between cytokine levels in the individual samples renders most of these differences not statistically significant.

11. *Fig. 8: CCl4 induced fibrosis: Axl identified as a driver of fibrosis as Axl^{-/-} mice show reduced fibrosis. Why is no reduction of fibrosis detected in Axl^{-/-}/Mertk^{-/-} mice?*

Reviewer 1 asks an interesting question. We do not know the answer, but have outlined two possibilities in the Discussion of the revised paper.

12. *Suppl Fig S2 is misleading. What are the differences between the two upper rows in panel A and C, both showing Axl expression in the control?*

There is no difference in the upper rows of panels A and C, except that the images in panel A are at a higher magnification. The bottom row in panel A is Mer, rather than Axl. The control images – upper rows in A and C, and lower row in A – illustrate that there is no overlap between Axl/Mer expression and the stellate cell marker Desmin in the absence of experimental challenge (in this case, CCl₄). In the revised paper, this figure is now Fig. S3.

13. *It should be shown whether Axl expression is induced in hepatocytes upon CCl4 treatment.*

Fig. S3 illustrates that there is no expression of Axl in hepatocytes, either before or after CCl₄.

14. *Authors should discuss data of human Axl and Mertk expression from patients with liver fibrosis.*

We have now addressed this question in the revised Discussion.

Reviewer #2

Zagórska, Través et al is a very interesting addition to our knowledge on the roles of AXL and MERTK in different pathological conditions in the liver. The manuscript provides a large amount of consistent data, revealing the specific role of each receptor... The histological characterization of the system is very conclusive.

We thank Reviewer 2 for this favorable evaluation of our study.

*Role of TAM receptors in the aging process of the liver: Aged *Axl*^{-/-} *Mertk*^{-/-} livers were damaged and inflamed in the absence of any experimental insult, suggesting TAM signaling is required for normal liver homeostasis and healthy aging. The data are supportive, however, if the authors could provide information from single *AXL*^{-/-} and *Mertk*^{-/-} aged mice it would emphasize the importance of the double suppression in age-related liver damage.*

As indicated above in response to the same point raised by Reviewer 1, we have now included new qPCR data on the levels of multiple cytokine and chemokine mRNAs in both *Axl*^{-/-} and *Mertk*^{-/-} mice.

*Role of TAM receptors in Jo2 and LPS/D-Gal acute injury models: In a fulminant hepatitis model based on injection of the Jo2 anti-Fas antibody only *Mertk*^{-/-} and *Axl*^{-/-} *Mertk*^{-/-} mice presented liver damaged and increased cleaved caspase-3. Minor point: Caspase 3 activation in the case of *Axl*^{-/-} animals (Fig 3B) seems intermediate compared to WT and *MerTK* mice. Is this correct?*

This does not appear to be the case, based on the quantification that we have performed in multiple mice: please see the results displayed in panel 3C.

*Similarly, in an endotoxic shock model after LPS and D-galactosamine injection, liver damaged and apoptotic cells were detected at 16 h only in *Mertk*^{-/-} and *Axl*^{-/-} *Mertk*^{-/-} mice, also exhibiting less recovery and regeneration after 7 days. The data are very convincing.*

Thank you.

The western showing AXL phosphorylation may suggest some increase, could you provide the quantification of AXL and MERTK p-Tyr in these experiments? In addition, have you checked for soluble AXL and MERTK serum levels after treatments?

As indicated in our response to the same point made by Reviewer 1 above, we have now provided quantification for these Fig. 3 p-Tyr results.

*TAM receptor expression in KCs is critical for liver physiology: Aged *Cx3cr1Cre/+Axl^{fl/fl}Mertk^{fl/fl}* livers displayed elevated levels of many proinflammatory and immunoregulatory mRNAs relative to *Axl^{fl/fl}Mertk^{fl/fl}*. The data are compelling.*

Thank you.

*Role of TAM receptors in acetaminophen-induced acute liver injury: The livers of APAP-treated mice at 48 hours after drug administration revealed massive liver damage specifically in the *Axl*^{-/-} mice. The data are supportive.*

Thank you.

*Do you have any insight of the mechanism involved in *Axl* activation by APAP? Could this be a redox-mediated mechanism?*

We have not investigated the mechanism directly. We have now included a brief

sentence highlighting the fact that oxidative stress has been shown to induce Axl expression and activation in several other systems (although we have not done experiments to address this specifically in our APAP studies), and have included two references that document this phenomenon in the revision.

Role of TAM receptors in response to lethal liver injury: A high (1 mg/kg) dose of Jo2 induced AXI and MERTK phosphorylation, increased sAXL levels. AXL^{-/-} and Mertk^{-/-} mice were more sensitive to Jo2 challenge relative to WT, as well as Tyro3^{-/-} Axl^{-/-} Mertk^{-/-} mice. The data are clear.

Thank you.

Are soluble levels of MERTK in serum increased after Jo2 administration? Is there a reason why TKO are shown whereas Axl^{-/-} Mertk^{-/-} are not?

We have not measured sMer in serum in the Jo2 models, since the production of sMer generally (in any setting) is much less robust than the production of sAxl. We have, however, examined steady-state levels of Mer and Axl in the liver at 2 hours after Jo2 treatment in new panel 7B. The generation of soluble versions of these receptors is always accompanied by reduced expression of the full-length receptors in tissues. This is very apparent for Axl in the 7B western blots, as expression of the full-length receptor is dramatically reduced at 2 hours after Jo2. In contrast, expression of full-length Mer is only modestly reduced by the same treatment (new 7B). We have also examined Mer (and Axl, Gas6, and F4/80) expression in the liver at 2 hours after a lethal Jo2 dose, by immunohistochemistry (IHC). We now include these (IHC) results in a new Supporting Fig. S2. Note that there is substantial loss of Axl and Gas6 (which is always complexed with Axl) in the liver 2 hours after Jo2 – consistent with Axl cleavage and the production of sAxl; but that there is still substantial – that is, nearly unchanged - Mer expression in the liver at the same time, consistent with minimal production of sMer.

There is no significant difference between the survival of TKOs versus Axl/Mer double mutants. We have therefore combined original panels 7D and E into a new panel 7D that displays survival curves for WT, Axl^{-/-}, Mertk^{-/-}, and Axl^{-/-}Mertk^{-/-} mice.

Role of TAM receptors in liver fibrosis: CCl4-driven hepatic fibrosis was specifically enhanced by Axl signaling and AXL^{-/-} mice protected. The data are strongly supportive.

Thank you.

Reviewer #3

The paper by Zagorska et al. summarises important data, which for the first time demonstrate the importance of constitutive TAM receptor Axl and Mertk expression on KCs for the maintenance of liver tissue homeostasis in the mouse. Their findings and conclusions are developed using a thorough methodology and convincing IHC/IF images, and confirmatory techniques.

We thank Reviewer 3 for this positive summary of our paper.

Main points

1. *The constitutive expression of Axl and Mer in KCs in adult mouse liver under physiological*

conditions was heretofore unknown and was clearly demonstrated by IHC (Fig 1). On page 11 the authors write "Most tissue macrophages express high and low levels of Mertk and Axl, respectively, ...". Please clarify or rephrase this statement. The image (Fig 1A, B and H) do not demonstrate the finding that Mertk is expressed at high and Axl at low levels on KC.

Indeed. Our point was that KCs are not like most tissue macrophages: they fall into the unusual group of macrophages that express high levels of both Mer and Axl. We mentioned red pulp macrophages as an additional example of cells that express high levels of both Mer and Axl, and included a reference. We have now clarified this statement, provided additional examples of both Mertk^{hi}Axl^{lo} and Mertk^{hi}Axl^{hi} macrophages, and also provided additional references.

2. The authors conclude that aged livers from Axl^{-/-}Mertk^{-/-} animals were "damaged and inflamed" (page 12) having shown an up-regulation of markers for apoptosis, macrophage activation and infiltration and other indirect markers. The findings should be supported by an H&E stain highlighting inflammation and possibly necrosis and fibrosis.

We have now included representative H&E images of the WT and Axl^{-/-}Mertk^{-/-} liver at 12 months of age in new Supporting Fig. 1A of the revised paper. These show an obvious increase in cellularity in the aged Axl^{-/-}Mertk^{-/-} liver, with what appear to be many infiltrating immune cells that are particularly prominent in the areas immediately surrounding large blood vessels.

3. The development of acute liver tissue injury in experimental models in TAM knockout mice is convincingly demonstrated and very interesting.

Thank you.

Fig 3B: in H&E the liver integrity looks better in Axl^{-/-}Mertk^{-/-} animals compared to Mertk^{-/-} animals. Is this just a matter of the picture chosen for the figure? In case this was a finding seen throughout the liver of all mice, are there any other pathways that may be compensatory up-regulated in Axl^{-/-}Mertk^{-/-} animals trying to maintain liver tissue homeostasis? Please review and append and/or comment. It is also interesting to see the recovery at day 7 - please add data for the Jo-2 model and include H&E in Fig 4F.

This is an interesting question that we have now addressed in the revised Discussion. With respect to cCasp3 there was no difference, but ALT and AST were marginally but not significantly worse in Mertk^{-/-} than in Axl^{-/-}Mertk^{-/-} mice. We do not have data for recovery at day 7 in the Jo2 model.

4. The conditional knockout mouse data supports the notion that TAM receptor expression in KC rather than LSEC is indeed controlling liver tissue homeostasis. It remains unclear why the role of Axl expression on peritoneal macrophages is stressed here and not mentioned in other sections of the manuscript.

We have clarified this point in the revised Results.

5. The data highlighting a role for Axl in tissue protection against APAP induced acute liver injury are interesting. It would be also interesting to see the extend of APAP induced liver injury in the Axl^{-/-}Mertk^{-/-} animals model. Moreover in respect to the controversy to the data reported in Ref 48, the expression of Axl and Mertk in liver tissue in the different strains should be shown.

The mice analyzed by Triantafyllou *et al.* were on a mixed genetic background, whereas our mice were pure C57Bl/6. That having been said, there really is not much of

a controversy, since the increase in liver necrosis reported by Triantafyllou *et al.* in the *Mertk*^{-/-} mice is very small (1.4-fold), and is only statistically significant at 8 and not at 24 hours post-treatment. We have therefore removed the discussion of this result from the Results, but mentioned it in passing in the revised Discussion.

6. It is most interesting to find that different models of acute liver injury seem to induce different pathways and involve different homeostasis mechanisms in KCs. The data and potential mechanisms behind this should be discussed in more detail or further explored. This is highly relevant in respect to potential future studies in humans and allows no conclusion for therapeutic translation to human diseases at this point.

We fully agree with Reviewer 3 on this point. We have added new discussion of this issue to the Discussion section of the paper.

7. In Fig 7A expression levels of Axl and Mertk should be shown. Regarding soluble Axl (Fig 7C): Also soluble Mertk should be included here. It should further be discussed, that their source does not consist of KCs only - but there are several other TAM receptor expressing cells in the body that may contribute to the soluble fraction including circulating monocytes.

Please see our response to Reviewer 2 above on this same point. We have added new data on Axl and Mer expression (and the possible production of sMer) in the liver at 2 hours after a lethal dose of Jo2 to both Fig. 7 and in new Supporting Fig. S2.

8. The expression of TAM receptors in the CCl4 model shown here is new and interesting but studied in less detail compared to the acute liver injury models. Data showing inflammation and KCs in different strains following CCl4 treatment are not shown. The bright Axl signal on IHC shown in Fig S2 are less convincing, the role of the EC is not addressed here, although part of the signal may be related to cells other than macrophages and HSC.

We have addressed this point in the revised Discussion. We do not exclude a role for Axl in HSCs or ECs.

Again, we are indebted to our referees for their detailed evaluation of our submission. We feel that the new data and new figures we have included in the revised manuscript, together with the amended and clarified presentation and discussion of the results, have significantly improved the paper.

May 25, 2020

RE: Life Science Alliance Manuscript #LSA-2020-00694R

Dr. Greg E. Lemke
Salk Institute for Biological Studies
10010 North Torrey Pines Road
La Jolla, CA 92037

Dear Dr. Lemke,

Thank you for submitting your revised manuscript entitled "Differential regulation of hepatic physiology and injury by the TAM receptors Axl and Mer". As you will see, the reviewers appreciate the introduced changes, and we would thus be happy to publish your paper in Life Science Alliance pending final revisions necessary to meet our formatting guidelines.

- Please mention the statistical test employed in each figure legend, next to the p-values
- Please make sure that the author names and author order are identical in our submission system when comparing to the manuscript docx file
- Please add the supplementary figure legends to the main manuscript file

A. FINAL FILES:

-- Summary blurb (enter in submission system): A short text summarizing in a single sentence the study (max. 200 characters including spaces). This text is used in conjunction with the titles of papers, hence should be informative and complementary to the title. It should describe the context and significance of the findings for a general readership; it should be written in the present tense

and refer to the work in the third person. Author names should not be mentioned.

B. MANUSCRIPT ORGANIZATION AND FORMATTING:

Sincerely,

Reilly Lorenz
Editorial Office Life Science Alliance
Meyerhofstr. 1
69117 Heidelberg, Germany
t +49 6221 8891 414
e contact@life-science-alliance.org
www.life-science-alliance.org

Reviewer #1 (Comments to the Authors (Required)):

I recommend the revised version of the manuscript for publication without further changes.

Reviewer #2 (Comments to the Authors (Required)):

The authors have successfully addressed the questions suggested by the reviewers. Thank you.

Reviewer #3 (Comments to the Authors (Required)):

The revised manuscript has been adequately changed and amended according to the questions and suggestions of the reviewers. I have no further comments.

May 29, 2020

RE: Life Science Alliance Manuscript #LSA-2020-00694RR

Dr. Greg E. Lemke
Salk Institute for Biological Studies
10010 North Torrey Pines Road
La Jolla, CA 92037

Dear Dr. Lemke,

Thank you for submitting your Research Article entitled "Differential regulation of hepatic physiology and injury by the TAM receptors Axl and Mer". It is a pleasure to let you know that your manuscript is now accepted for publication in Life Science Alliance. Congratulations on this interesting work.

DISTRIBUTION OF MATERIALS:

Again, congratulations on a very nice paper. I hope you found the review process to be constructive and are pleased with how the manuscript was handled editorially. We look forward to future exciting submissions from your lab.

Sincerely,

Reilly Lorenz
Editorial Office Life Science Alliance
Meyerhofstr. 1
69117 Heidelberg, Germany
t +49 6221 8891 414
e contact@life-science-alliance.org
www.life-science-alliance.org